# Photobody formation spatially segregates two opposing phytochrome B signaling actions of PIF5 degradation and stabilization

Ruth Jean Ae Kim [1,3], De Fan [1,3], Jiangman He [1,3], Keunhwa Kim[1,2], Juan Du [1] & Meng Chen [1] ✉

Photoactivation of the plant photoreceptor and thermosensor phytochrome B (PHYB) triggers its condensation into subnuclear membraneless organelles named photobodies (PBs). However, the function of PBs in PHYB signaling remains frustratingly elusive. Here, we found that PHYB recruits PHYTOCHROME-INTERACTING FACTOR 5 (PIF5) to PBs. Surprisingly, PHYB exerts opposing roles in degrading and stabilizing PIF5. Perturbing PB size by overproducing PHYB provoked a biphasic PIF5 response: while a moderate increase in PHYB enhanced PIF5 degradation, further elevating the PHYB level stabilized PIF5 by retaining more of it in enlarged PBs. Conversely, reducing PB size by dim light, which enhanced PB dynamics and nucleoplasmic PHYB and PIF5, switched the balance towards PIF5 degradation. Together, these results reveal that PB formation spatially segregates two antagonistic PHYB signaling actions – PIF5 stabilization in PBs and PIF5 degradation in the surrounding nucleoplasm – which could enable an environmentally sensitive, counter-balancing mechanism to titrate nucleoplasmic PIF5 and environmental responses.

A universal architectural hallmark of the cell nucleus is various functionally distinct subnuclear membraneless organelles, collectively referred to as nuclear bodies[1–3]. Nuclear bodies act as liquid- or gel-like biomolecular condensates that are commonly thought to be excluded from the surrounding nucleoplasm via concentration-dependent, energetically favorable liquid-liquid phase separation (LLPS)[3–5]. Many nuclear bodies, such as nucleoli, Cajal bodies, nuclear speckles, paraspeckles, histone locus bodies, and promyelocytic leukemia protein (PML) bodies, are well characterized, and they are associated with diverse basic nuclear functions in transcription, splicing, and DNA replication and repair. Accumulating observations have also documented the nonuniform subnuclear distribution of cell signaling molecules to nuclear bodies, in addition to the surrounding nucleoplasm. For instance, the tumor suppressor p53 is recruited to PML bodies[6]. However, the functional significance of the spatial segregation of signaling molecules into two phase-separated compartments – nuclear bodies and the surrounding nucleoplasm – remains elusive. A better understanding of the function of nuclear bodies in cell signaling requires genetically tractable models directly connecting nuclear body dynamics to cell signaling. Photobodies (PBs) are plant nuclear bodies defined molecularly by the presence of the photoreceptor and thermosensor phytochrome B (PHYB)[7–9]. Environmental light and temperature cues control PHYB activity and, therefore, regulate the assembly/dissolution of PBs and PHYB signaling outputs[7,10,11]. As such, PBs provide a unique experimental paradigm for interrogating the general principles of nuclear bodies in cell signaling.

PHYs are evolutionarily conserved red (R) and far-red (FR) photoreceptors in plants[12]. The prototypical plant PHY is a homodimer; each monomer contains an N-terminal photosensory module and a C-terminal output module[12,13]. PHYs can be photoconverted between

[1]Department of Botany and Plant Sciences, Institute for Integrative Genome Biology, University of California, Riverside, CA 92521, USA. [2]Present address: Plant Molecular Biology and Biotechnology Research Center, Gyeongsang National University, Jinju 52828, Republic of Korea. [3]These authors contributed equally: Ruth Jean Ae Kim, De Fan, Jiangman He. ✉e-mail: meng.chen@ucr.edu

two relatively stable forms: the R-light-absorbing inactive Pr form and the FR-light-absorbing active Pfr form[12,13]. The genome of the reference plant species *Arabidopsis thaliana* (*Arabidopsis*) encodes five PHYs, PHYA-E, among which PHYB plays a prominent role[14]. The photoconversion of PHYB – either photoactivation by R light or photoinhibition by FR light – alters the Pfr/Pr equilibrium of individual PHYB molecules and the steady-state cellular concentration of the active PfrPfr homodimer, thereby allowing plants to monitor changes in light quality, quantity, and periodicity. The amount of active PHYB can also be modulated by ambient temperature through temperature-dependent thermal reversion from Pfr to Pr[15], making PHYB a thermosensor in addition to a photoreceptor[11,16]. By controlling the amount of active PHYB, environmental light and temperature cues regulate all aspects of plant development and growth, including germination, seedling establishment, shade avoidance, floral induction, senescence, and immunity[17,18]. PHYB signaling is best studied during *Arabidopsis* seedling establishment, where the perception of environmental cues by PHYB in the epidermal cells of the embryonic leaves (cotyledons) controls the biosynthesis of the mobile growth hormone auxin to regulate seedling morphogenesis, including promoting cotyledon expansion locally, as well as inhibiting the elongation of the embryonic stem (hypocotyl) remotely[17,18].

The PHYB-containing subnuclear membraneless organelle was named the "photobody" by Joanne Chory in reference to its dynamic assembly and dissolution in response to R and FR light[9]. Although "photobody" was also adopted to describe the fascinating blue-light-inducible cryptochrome 2 (CRY2)-containing nuclear bodies[19], colocalization studies using BY-2 and *Arabidopsis* protoplasts suggest that the PHYB-containing PBs and the CRY2-containing condensates may represent distinct nuclear bodies[20,21]. The regulation of PBs by R/FR light and temperature has been extensively studied in *Arabidopsis* pavement epidermal cells using fluorescent-protein-tagged PHYB (PHYB-FP). The current model posits that PB formation and maintenance are driven by the condensation of the active PfrPfr form of PHYB. This conclusion is supported by several lines of evidence. First, PB formation and dissolution can be induced by the activation and inactivation of PHYB, respectively. Akira Nagatani, Ferenc Nagy, and Eberhard Schäfer first reported that the photoactivation of PHYB-FP triggers its nuclear accumulation and further compartmentalization into discrete subnuclear speckles (PBs)[22,23]. Conversely, during the light-to-dark transition or upon an FR treatment where PHYB reverts to the inactive form, PHYB-FP moves from PBs into the nucleoplasm[24–27]. Second, the steady-state pattern of PBs is dependent on light intensity and correlates with the amount of active PHYB[28]. Under strong R light (e.g., 10 μmol m$^{-2}$ s$^{-1}$), where each PHYB molecule stays in the Pfr form at least 50% of the time, PHYB-FP assembles into two to ten large PBs of 0.7–2 μm in diameter[10,24,28,29]. In contrast, under dim R light, where PHYB stays as the Pr form most of the time, PHYB-FP localizes to tens of small PBs of 0.1–0.7 μm in diameter, or it is dispersed evenly in the nucleoplasm[10,24,28]. Consistent with these observations, a constitutively active *phyB* mutant YHB, which carries a Y276H mutation in PHYB's chromophore attachment domain and locks PHYB in an active form, localizes to a few large PBs even in the dark[30]. In contrast, *phyB* mutants that destabilize the Pfr form – e.g., removing the disordered N-terminal extension[15,31,32] – promote PB dissolution[28,31–33]. Third, PB formation relies on the dimerization of PHYB's C-terminal module[32,34,35]. A D1040V mutation in this module, which disrupts PHYB dimerization, abolishes PHYB's ability to form PBs[36]. Fourth, PHYB can undergo light-dependent LLPS into biomolecular condensates in mammalian cells[37,38]. The intramolecular attributes of PHYB required for PB formation in vivo, e.g., the C-terminal module and N-terminal extension, are also essential for PHYB LLPS in this heterologous system[37]. These observations support the idea that PB formation in vivo may be propelled by the LLPS of active PHYB alone, though PHYB condensation under physiological conditions is likely more complex involving other

molecules such as PHOTOPERIODIC CONTROL OF HYPOCOTYL 1 (PCH1)[26,39].

PHYB controls diverse developmental responses primarily by regulating a family of basic helix-loop-helix transcription factors called PHYTOCHROME-INTERACTING FACTORs (PIFs)[18,40]. In general, PIFs are antagonists of PHYB signaling. For instance, the five prominent PIFs – PIF1, PIF3, PIF4, PIF5, and PIF7 – collectively promote hypocotyl elongation[18,40]. A central mechanism of PHYB signaling is to inhibit the function of PIFs, including promoting their degradation[18,40], attenuating their DNA binding[41,42], and repressing their transactivation activity[43]. PIF1, PIF3, PIF4, and PIF5 are rapidly degraded during the dark-to-light transition in a PHYB-dependent manner[44–48]. PHYB binds directly to the prototypical PIF, PIF3, to promote its phosphorylation by PHOTOREGULATORY PROTEIN KINASEs (PPKs) and subsequent ubiquitin-proteasome-dependent degradation[44,49–51]. Two E3 ubiquitin ligases can ubiquitylate PIF3, including the **c**ullin3-**R**ING E3 ubiquitin **l**igases (CRL) with LIGHT-RESPONSE BRIC-A-BRAC/TRAMTRACK/BROADs (LRBs) as the substrate recognition subunits (CRL3$^{LRB}$) and the cullin1-RING E3 with EIN3-BINDING F BOX PROTEINs (EBFs) as the substrate recognition subunits (CRL1$^{EBF}$)[50,51]. In addition to regulating PIFs via direct interaction, PHYB also promotes PIF degradation indirectly by inhibiting factors that stabilize PIFs, such as the CONSTITUTIVE PHOTOMORPHOGENIC 1 (COP1) and SUPPRESSOR OF PHYTOCHROME A-105 proteins (SPAs), which constitute the substrate recognition subunits of the CRL4$^{COP1/SPA}$ E3 ubiquitin ligases[52–54]. Interestingly, although PIF1, PIF3, PIF4, and PIF5 are subject to PHYB-mediated degradation, they exhibit different accumulation patterns. While PIF1 and PIF3 accumulate only in darkness and become undetectable in the light[29,36], PIF4 and PIF5 can accumulate significantly under prolonged light treatment[36,46,48,55]. The accumulation of PIF4 and PIF5 in the light plays an essential role in plants' responses to warm temperatures during the daytime[55,56]. However, the mechanism stabilizing PIF4 and PIF5 in the light remains elusive.

PBs are considered to be associated with PHYB signaling[7,8]. PB formation correlates with PHYB-mediated light responses, such as the inhibition of hypocotyl elongation[24,28,29]. *phyB* mutants defective in PB formation are impaired in PHYB signaling[28,30–34]. Recent proteomics studies revealed that many PHYB signaling components reside in PBs; these components include COP1, SPAs, PPKs, PCH1, PCH1-LIKE (PCHL), and transcription regulators in light and photoperiodic signaling, including PIF4, TANDEM ZINC-FINGER-PLUS 3 (TZP), EARLY FLOWERING 3 (ELF3), EARLY FLOWERING 4 (ELF4), and LUX ARRHYTHMO (LUX), lending further support to the functional role of PBs in PHYB signaling[21,26,57]. The current data support the model that PBs play a vital role in PIF3 degradation. During the dark-to-light transition, FP-tagged PIF3 localizes to PBs before its degradation[44,58]. Conversely, PIF3 reaccumulation during the light-to-dark transition correlates with the disappearance of PBs[24]. Forward genetic screens have identified three PB-deficient mutants, *hmr* (*hemera*)[29,59,60], *rcb* (*regulator of chloroplast biogenesis*)[61], and *ncp* (*nuclear control of pep activity*)[62], in which PHYB-FP fails to form large PBs and PIF3 degradation is blocked in the light. Moreover, overexpressing PHYB's C-terminal output module alone, which localizes constitutively to PBs, can mediate PIF3 degradation even in darkness[36].

Despite the accumulating evidence associating PBs with PIF3 degradation, the precise function of PBs remains frustratingly elusive. We still cannot unequivocally conclude that PBs are the sites of PIF3 degradation, and it is still unknown whether PBs also regulate the stability of other PIFs. A significant challenge in dissecting the function of PBs, and also nuclear bodies in general, has been the difficulty of uncoupling the functional output of the nuclear-body comparment from that of the surrounding nucleoplasm. Because components can diffuse between nuclear bodies and the surrounding nucleoplasm, although manipulating those components may disrupt nuclear body assembly and the functional output associated with the components,

such a correlation is usually insufficient to assign the functional defects exclusively to the nuclear body compartment. The previous approach of using loss-of-function mutants in PHYB or PHYB signaling could disrupt the PHYB signaling outputs of either or both PBs and the surrounding nucleoplasm and, therefore, could not uncouple the signaling actions of the two phase-separated compartments. Another hurdle in studying PIF3 degradation was the difficulty in monitoring its PB localization. Because PIF3 does not accumulate to a detectable level in the light, FP-tagged PIF3 can only be observed briefly during the dark-to-light transition[44,58].

To circumvent these obstacles to characterizing PBs, we implemented two major strategic changes in the current study. First, we adopted PIF5 as a model because PIF5 accumulates in the light and could potentially allow us to visualize its localization to PBs in light-grown seedlings[36,46,48,55], though the PB localization of PIF5 had not been reported previously. Second, instead of disrupting PBs using loss-of-function mutants in PHYB or PHYB signaling, we perturbed the PB size by increasing PHYB abundance. These approaches show that PHYB recruits PIF5 to PBs and, surprisingly, that PHYB exerts two opposing functions in degrading and stabilizing PIF5. Our results reveal that PB formation allows the phase separation and competition of the two antagonistic PHYB signaling actions – PIF5 stabilization in PBs and PIF5 degradation in the nucleoplasm. We propose a PB-enabled counterbalancing mechanism to titrate nucleoplasmic PIF5 and its signaling outputs.

## Results

### Increasing PHYB production alters PB size and dynamics

Instead of interrogating the function of PBs in loss-of-function phyB or PHYB signaling mutants, we explored an alternative approach to perturb PBs by increasing PHYB abundance. The current hypothesis is that PBs form via the LLPS of PHYB[37]. Based on the theory of the LLPS, the LLPS of PHYB occurs when PHYB accumulates at a critical concentration. The PHYB LLPS model predicts that the overproduction of PHYB above the critical concentration is expected to result only in the growth of PBs without changing the concentration of PHYB in either the PB or the surrounding nucleoplasmic compartment[4,63]. Therefore, if the model were correct, increasing the PHYB level should only enhance the signaling output from PBs while leaving the functional output of the nucleoplasmic PHYB unchanged or less affected. To test this hypothesis, we collected Arabidopsis lines expressing various amounts of PHYB. We previously reported the PBC line, which over-expresses CFP-tagged PHYB (PHYB-CFP) in the phyB-9 background[32]. PBC exhibits a short hypocotyl phenotype, as it overexpresses PHYB-CFP at a level 65-fold that of the endogenous PHYB (Fig. 1a–c)[32]. To create lines expressing intermediate levels of PHYB between PBC and Col-0, we generated new transgenic lines in the phyB-9 background carrying PHYB genomic DNA with a CFP sequence inserted immediately before the PHYB stop codon. We named them gPBC (genomic PHYB-CFP) lines. Two single-insertion gPBC lines, gPBC-25 and gPBC-29, were selected for further analysis. The steady-state levels of PHYB-CFP in gPBC-25 and gPBC-29 were about 7- and 40-fold that of the endogenous PHYB level in Col-0, respectively (Fig. 1a–c). Both gPBC lines rescued the long-hypocotyl phenotype of phyB-9, indicating that PHYB-CFP was functional (Fig. 1a, b). Corroborating the notion that the PHYB response correlates with the PHYB level, the hypocotyl length of gPBC-25 was in between that of Col-0 and PBC, whereas the hypocotyl length of gPBC-29 was similar to that of PBC (Fig. 1a, b). With the two new gPBC lines, we had a panel of five lines – phyB-9, Col-0, gPBC-25, gPBC-29, and PBC – with PHYB levels ranging from zero to 65-fold that of the wild-type level (Fig. 1a–c).

To test whether increasing the PHYB level enhances the size of PBs, we measured the volumes of PHYB-CFP PBs in gPBC-25, gPBC-29, and PBC. Indeed, the PB size increased with the PHYB-CFP level (Fig. 1d, e). PBC had the largest PBs, which were, on average, more than

five-fold larger than those in gPBC-25. The average PB volume of gPBC-29 was three times larger than that of gPBC-25 (Fig. 1e). The fraction of PHYB-CFP localized to PBs also increased with the PHYB-CFP level (Fig. 1f). These results support the predictions of the PHYB LLPS model that increasing PHYB abundance enlarges PBs. However, these data did not assess whether the concentration of PHYB-CFP remained the same in the PBs and the surrounding nucleoplasm. We used fluorescence recovery after photobleaching (FRAP) to evaluate the exchange of PHYB-CFP molecules between PBs and the surrounding nucleoplasm. To our surprise, the PBs from gPBC-25 and PBC exhibited significant differences in the dynamics of PHYB-CFP. The fluorescence recovery in gPBC-25 was only 29.9%, suggesting that the majority of PHYB-CFP molecules were not mobile (Fig. 1g). The percentage of fluorescence recovery in PBC was further reduced to 19.4%, indicating that increasing PHYB-CFP abundance decreased the mobile fraction of PHYB-CFP in PBs, likely due to a transition to a gel-like state (Fig. 1g). Supporting this idea, PHYB-CFP in the PBs from PBC showed a decrease in the fluorescence recovery kinetics compared with gPBC-25, indicating a reduction in the diffusion rate of PHYB-CFP in PBC (Fig. 1g). Together, these results demonstrate that increasing PHYB abundance enhances PB size by recruiting a larger fraction of PHYB to the PB compartment and also induces a transition of PBs to a gel-like state that could retain PHYB for a longer time in PBs. Thus, this panel of lines with various PB sizes and PHYB-CFP dynamics provides an opportunity to interrogate the function of PBs in PIF degradation.

### PIF5 is a short-lived protein localized in PBs

To investigate the function of PBs in PIF regulation, we turned to PIF5 as a model because, despite PHYB-mediated PIF5 degradation[45,46,48], PIF5 can accumulate in the light[36,46,48,55,64], which could potentially allow us to monitor its subnuclear localization in the light. We generated transgenic lines expressing PIF5 fused with Myc and mCherry to the N-terminus (mCherry-PIF5) under the native PIF5 promoter in the null pif5-3 mutant background. Two independent transgenic lines with a single insertion of the transgene, named mCherry-PIF5/pif5-3 #8 and #9, were selected for further analysis. The pif5-3 mutant exhibited a short hypocotyl phenotype at 16 °C. Because PIF5 is required for the warm temperature-induced hypocotyl elongation, the defect of pif5-3 in hypocotyl growth became more pronounced at 27 °C (Fig. 2a, b)[55]. The mCherry-PIF5/pif5-3 lines rescued the hypocotyl-growth defects of pif5-3 at both temperatures (Fig. 2a, b), indicating that mCherry-PIF5 was functional. Similar to endogenous PIF5, mCherry-PIF5 accumulated in the light and could be detected by immunoblotting using anti-PIF5 antibodies (Fig. 2c). Despite being controlled by the native PIF5 promoter, the steady-state levels of mCherry-PIF5 in the transgenic lines were higher than that of endogenous PIF5 in Col-0. The mCherry-PIF5 level in line #8 was more than five-fold that of line #9 and more than ten-fold that of endogenous PIF5 in Col-0 (Fig. 2c). However, to our surprise, no mCherry signal could be detected by confocal microscopy in either of the mCherry-PIF5/pif5-3 transgenic lines. We reasoned that this discrepancy might be attributable to a faster turnover rate of mCherry-PIF5 compared with the maturation time required for newly synthesized mCherry to become fluorescent. The reported maturation time of mCherry is more than 60 min[65]. If the half-life of mCherry-PIF5 is significantly shorter than 60 min, the newly synthesized mCherry-PIF5 would have been degraded before becoming fluorescent. If this were the case, the non-fluorescent mCherry-PIF5 protein should be detectable by immunolocalization. To test the hypothesis, we performed immunofluorescence staining using anti-Myc antibodies. Indeed, Myc-tagged mCherry-PIF5 could be detected by immunostaining. More interestingly, mCherry-PIF5 was localized to discrete punctate structures at 16 °C and 27 °C (Fig. 2d). Simultaneously labeling endogenous PHYB using anti-PHYB antibodies demonstrated that mCherry-PIF5 colocalized with PHYB in PBs (Fig. 2d, e).

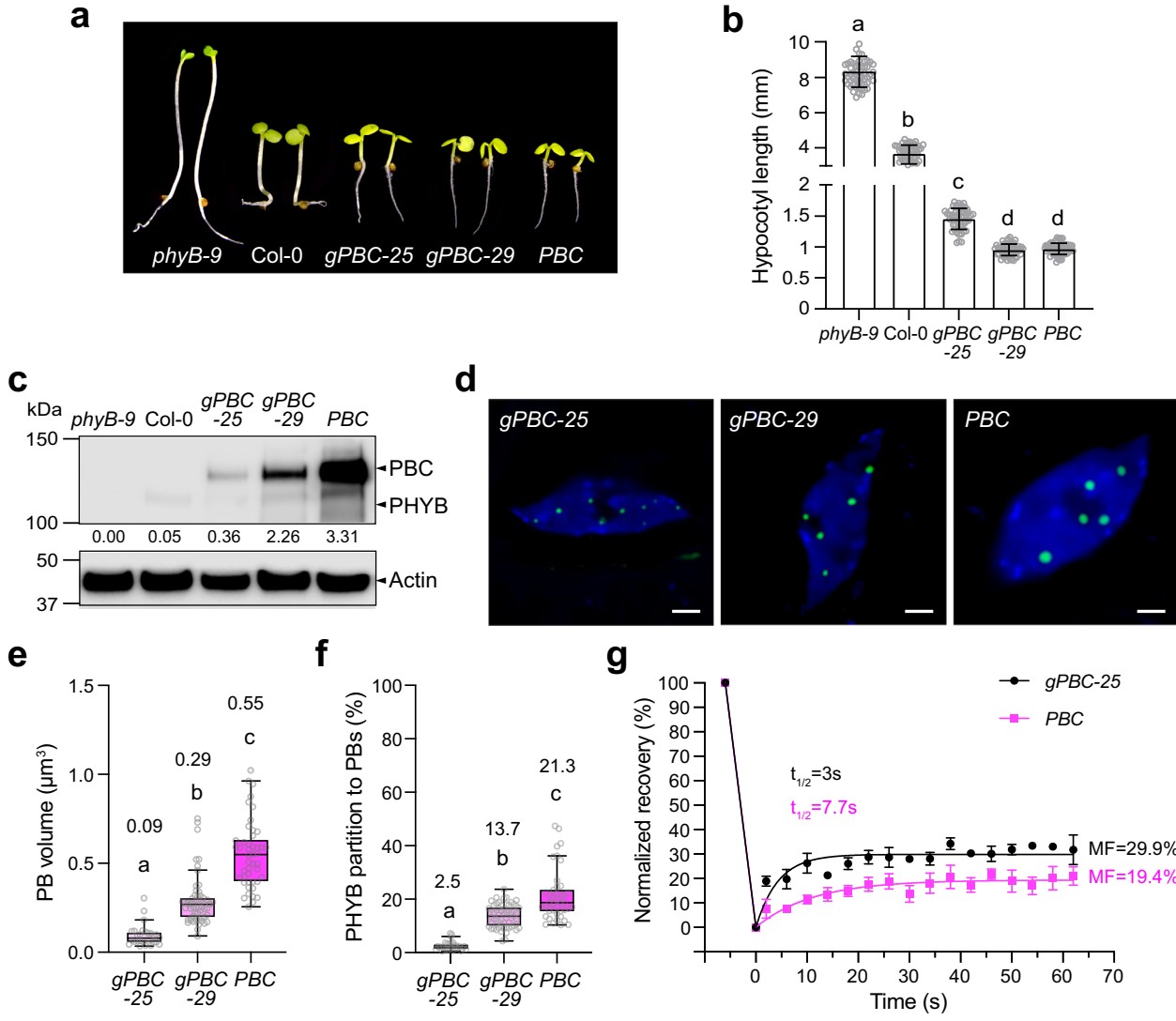

**Fig. 1 | Increasing PHYB abundance alters PB size and dynamics. a** Images of 4-d-old *phyB-9*, Col-0, *gPBC-25*, *gPBC-29* and *PBC* seedlings grown under 10 μmol m⁻² s⁻¹ R light at 21 °C. **b** Hypocotyl length measurements of the seedlings shown in (**a**). Error bars represent the s.d. (*n* > 50 seedlings); the centers of the error bars indicate the mean. Different letters denote statistically significant differences in hypocotyl length (ANOVA, Tukey's HSD, multiplicity adjusted $p \leq 0.05$). **c** Immunoblots showing the PHYB levels in the seedlings described in (**a**). Actin was used as a loading control. The relative endogenous PHYB and PHYB-CFP levels normalized to the corresponding levels of actin are shown. The immunoblot experiments were independently repeated three times with similar results. **d** Confocal images showing representative PHYB-CFP PB patterns in cotyledon pavement epidermal cells from 4-d-old *gPBC-25, gPBC-29* and *PBC* seedlings grown under 10 μmol m⁻² s⁻¹ R light at 21 °C. Scale bars are equal to 2 μm. **e** Quantification of the PB volume in cotyledon epidermal cells from *gPBC-25, gPBC-29* and *PBC* as shown in (**d**). Different letters denote statistically significant differences in volume (ANOVA, Tukey's HSD,

multiplicity adjusted $p < 0.05$). **f** Quantification of the percentage of PHYB partitioned to PBs based on the total PHYB signal within the nuclei of *gPBC-25, gPBC-29* and *PBC* as shown in (**d**). Box and whisker plots showing the percent of PHYB partitioned to PBs per nucleus. Different letters denote statistically significant differences in percentage (ANOVA, Tukey's HSD, multiplicity adjusted $p < 0.05$). For (**e**, **f**), the numbers indicate the mean values. In the box and whisker plots, the boxes represent the 25th to 75th percentiles, and the bars are equal to the median. **g** Results of FRAP experiments showing normalized fluorescence recovery plotted over time for PHYB-CFP in PBs of pavement cell nuclei from 4-d-old *gPBC-25* (black) and *PBC* (magenta) seedlings grown under 50 μmol m⁻² s⁻¹ R light. Values represent the mean, and error bars represent the s.e. of the mean of three biological replicates. The solid lines represent an exponential fit of the data. MF: mobile fraction; $t_{1/2}$ represents the half-time of fluorescence recovery. The source data underlying the hypocotyl measurements in (**b**), the immunoblots in (**c**), and the PB characterization in (**e**–**g**) are provided in the Source Data file.

Thus, these results indicate that mCherry-PIF5 is a short-lived protein that colocalizes with PHYB in PBs under the physiological PHYB concentration.

**PHYB signaling both degrades and stabilizes PIF5**

The fact that mCherry-PIF5 was short-lived and localized in PBs might suggest that mCherry-PIF5 degradation occurs in PBs. However, it was equally possible that mCherry-PIF5 was degraded in the nucleoplasm and that because mCherry-PIF5 was exchanged between the nucleoplasm and PBs, both the nucleoplasmic and PB

pools of mCherry-PIF5 were short-lived. One way to distinguish between these two possibilities would be to test whether perturbing PBs alters PIF5 degradation. If PIF5 were degraded in PBs, increasing the size of PBs should recruit more PIF5 to PBs and accelerate PIF5 degradation. To test the hypothesis, we examined the endogenous levels of PIF5 in the panel of lines expressing various amounts of PHYB (Fig. 1a–c). Surprisingly, PIF5 showed a biphasic response to increases in PHYB abundance. In the range of zero to moderate levels of PHYB in *phyB-9*, Col-0, and *gPBC-25*, the steady-state level of PIF5 decreased with the increases in PHYB (Fig. 3a–c), supporting the

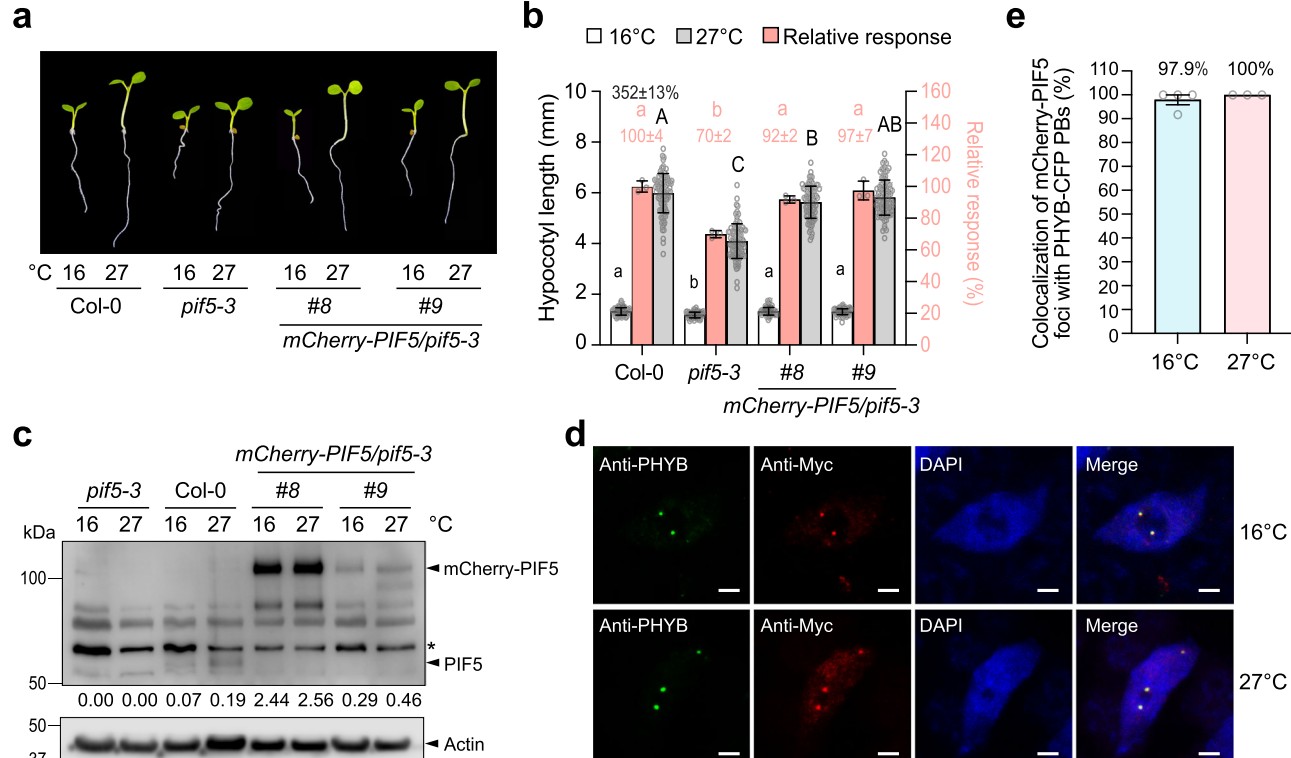

**Fig. 2 | mCherry-PIF5 is a short-lived protein localized in PBs. a** Images of 4-d-old seedlings of Col-0, *pif5-3* and two *mCherry-PIF5/pif5-3* (#8 and #9) lines grown under 50 μmol m$^{-2}$ s$^{-1}$ R light at 16 °C or 27 °C. **b** Hypocotyl length measurements of the seedlings described in **a**. The open and gray bars represent hypocotyl length measurements at 16 °C and 27 °C, respectively. Error bars for the hypocotyl measurements represent the s.d. ($n = 90$ seedlings). Lowercase and uppercase letters denote statistically significant differences in hypocotyl lengths at 16 °C or 27 °C, respectively (ANOVA, Tukey's HSD, multiplicity adjusted $p < 0.05$, $n = 90$ seedlings). The black number above the Col-0 data represents the percent increase in hypocotyl length at 27 °C compared to 16 °C (mean ± s.d., $n = 90$ seedlings). The pink bars show the relative response, defined as the hypocotyl response to 27 °C of a mutant or transgenic line relative to that of Col-0 (set at 100%). Error bars for the relative responses represent the s.e. of three biological replicates. The centers of all error bars indicate the mean. Pink numbers show the mean ± s.e. of the relative responses. Different pink letters denote statistically significant differences in the relative responses (ANOVA, Tukey's HSD, multiplicity adjusted $p < 0.05$, $n = 3$

biological replicates). **c** Immunoblots showing the steady-state levels of PIF5 and mCherry-PIF5 in 4-d-old Col-0, *pif5-3*, and the *mCherry-PIF5* lines (#8, #9) grown under 50 μmol m$^{-2}$ s$^{-1}$ R light at 16 °C or 27 °C. PIF5 and mCherry-PIF5 were detected using anti-PIF5 antibodies. Actin was used as a loading control. The relative levels of PIF5 or mCherry-PIF5 normalized to actin are shown. The asterisk indicates a nonspecific band. **d** Confocal images showing the colocalization of mCherry-PIF5 and PHYB in PBs in cotyledon pavement epidermal nuclei from 4-d-old *mCherry-PIF5* line #8 seedlings grown under 50 μmol m$^{-2}$ s$^{-1}$ R light at 16 °C or 27 °C. PHYB (green) and Myc-tagged *mCherry-PIF5* (red) were labeled via immunofluorescence staining using anti-PHYB and anti-Myc antibodies, respectively. Nuclei were stained with DAPI (blue). Scale bars are equal to 2 μm. **e** Quantification of the colocalization of mCherry-PIF5 foci with PHYB-CFP PBs in the experiments described in (**d**). Error bars represent the s.e. of at least three biological replicates. The source data underlying the hypocotyl measurements in (**b**), the immunoblots in (**c**), and the PB characterization in (**e**) are provided in the Source Data file.

current model that PHYB promotes PIF5 degradation in the light. However, further increasing PHYB abundance in *gPBC-29* and *PBC* unexpectedly enhanced PIF5 accumulation (Fig. 3a, b). Despite the changes in the PHYB level, the transcript levels of *PIF5* remained the same in *phyB-9*, Col-0, *gPBC-25*, *gPBC-29*, and *PBC* (Fig. 3c). Therefore, the changes in the protein level of PIF5 must be due to either PIF5 translation or degradation. We next examined the degradation kinetics of PIF5 in Col-0, *gPBC-25*, and *PBC* by treating the cell lysates with the translation inhibitor cycloheximide and then monitoring the disappearance of PIF5 over time. PIF5 was degraded much faster in *gPBC-25* than in Col-0 (Fig. 3d, e). Surprisingly, however, PIF5 degradation was attenuated dramatically in *PBC*. Thus, the accumulation of PIF5 in *PBC* and *gPBC-29* was most likely due to the PHYB-mediated stabilization of PIF5. These results suggest that PHYB signaling exerts opposing roles in both degrading and stabilizing PIF5, the balance of which can be adjusted by altering the PHYB level. Because the size of PBs increased significantly from *gPBC-25* to *PBC* (Fig. 1d, e), these results raised the hypothesis that while PHYB promotes PIF5 degradation in the nucleoplasm, it stabilizes PIF5 in PBs. As such, increasing the PB size in *PBC* could switch the balance towards PIF5 stabilization by recruiting more PIF5 to PBs.

## PHYB recruits and stabilizes PIF5 in PBs

To further examine the role of PBs in stabilizing PIF5, we reasoned that if PIF5 were stabilized due to its enhanced recruitment to the enlarged PBs in *PBC*, we might be able to observe the accumulation of stabilized or longer-lived mCherry-PIF5 proteins in PBs in *PBC* via confocal microscopy. To that end, we generated transgenic lines expressing mCherry-PIF5 tagged with human influenza hemagglutinin (HA) (HA-mCherry-PIF5) under the constitutive *UBIQUITIN 10* promoter in *PBC* (*mCherry-PIF5/PBC*). We selected two *mCherry-PIF5/PBC* lines (#1 and #9) for further analysis. The *mCherry-PIF5/PBC* lines were taller than *PBC* at 16 °C and 27 °C, suggesting that the HA-tagged mCherry-PIF5 was functional (Fig. 4a, b). As endogenous PIF5, mCherry-PIF5 could accumulate in both *mCherry-PIF5/PBC* lines in the light at 16 °C and 27 °C (Fig. 4c). Similar to the *mCherry-PIF5/pif5-3* lines (Fig. 3c), the mCherry-PIF5 levels in the *mCherry-PIF5/PBC* lines were more than ten-fold greater than that of the endogenous PIF5 in Col-0 (Fig. 4c). However, unlike the *mCherry-PIF5/pif5-3* lines, we could observe the fluorescent signal of mCherry-PIF5 in the *mCherry-PIF5/PBC* lines using confocal microscopy (Fig. 4d). mCherry-PIF5 was also colocalized with PHYB-CFP in PBs in the *mCherry-PIF5/PBC* lines at both 16 °C and 27 °C in the light (Fig. 4d, e). In dark-grown *mCherry-PIF5/PBC* seedlings

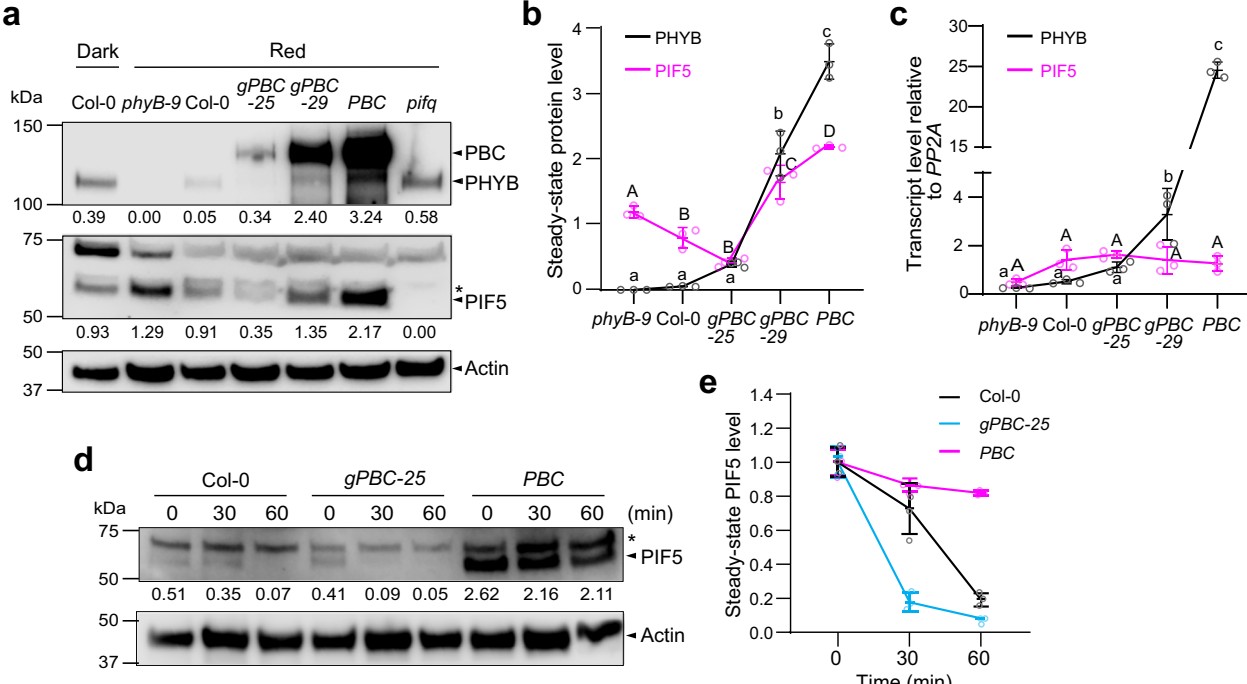

**Fig. 3 | Increasing PHYB abundance provokes a biphasic PIF5 response.**
**a** Immunoblots showing PHYB and PIF5 levels in 4-d-old *phyB-9*, Col-0, *gPBC-25*, *gPBC-29* and *PBC* seedlings grown under 10 μmol m⁻² s⁻¹ R light at 21 °C. 4-d-old dark-grown Col-0 and R-light-grown *pifq* were positive and negative controls for PIF5, respectively. The immunoblot experiments were independently repeated three times with similar results. **b** Quantification of the PHYB and PIF5 levels shown in (**a**). Error bars represent the s.d. of three independent replicates. The centers of the error bars indicate the mean values. Different lowercase and uppercase letters denote statistically significant differences in the abundance of PHYB and PIF5, respectively (ANOVA, Tukey's HSD, multiplicity adjusted $p < 0.05$, $n = 3$ biological replicates). **c** Quantitative real-time PCR (qRT-PCR) analysis of the transcript levels of *PHYB* and *PIF5* in seedlings shown in (**a**). Different lowercase and uppercase

letters denote statistically significant differences in *PHYB* and *PIF5* transcripts, respectively (ANOVA, Tukey's HSD, multiplicity adjusted $p < 0.05$, $n = 3$ biological replicates). **d** Immunoblots showing the degradation kinetics of PIF5 in Col-0, *gPBC-25*, and *PBC*. 4-d-old Col-0, *gPBC-25*, and *PBC* seedlings were incubated with cycloheximide and collected at the indicated time points. Total proteins were extracted and subjected to immunoblotting analysis using anti-PIF5 antibodies. For (**a**, **d**), actin was used as a loading control. The asterisk indicates a nonspecific band. The relative PHYB or PIF5 levels normalized to the corresponding actin levels are shown. **e** Quantification of the relative PIF5 protein levels shown in (**d**). Error bars represent the s.d. of three independent replicates. The source data underlying the immunoblots in (**a**, **b**, **d**, **e**), and the qRT-PCR analysis in (**c**) are provided in the Source Data file.

where there were no PHYB-CFP PBs, mCherry-PIF5 was evenly dispersed in the nucleoplasm (Fig. 4f), implying that mCherry-PIF5 was recruited to PBs in the light by PHYB. The fact that mCherry-PIF5 became detectable by confocal microscopy suggested that the half-life of the mCherry-PIF5 in *mCherry-PIF5/PBC* was significantly longer than that in *mCherry-PIF5/pif5-3*. Supporting this conclusion, mCherry-PIF5 accumulated to higher levels in *mCherry-PIF5/PBC* than *mCherry-PIF5/pif5-3* (Figs. 3a, b and 4c). Moreover, the degradation kinetics of mCherry-PIF5 in *mCherry-PIF5/PBC* became slower than that in *mCherry-PIF5/pif5-3* (Fig. 4g, h). It is important to note that the degradation kinetics of mCherry-PIF5 (Fig. 4g, h) was considerably faster than that of endogenous PIF5 (Fig. 3d, e)[48], this could be attributed to either the mCherry tag or the overexpression of mCherry-PIF5 and the resulted change in mCherry-PIF5's PB/nucleoplasm partitioning. Together, these results support the model that PHYB recruits and stabilizes PIF5 in PBs and that the action of PHYB in PIF5 stabilization is enhanced in *PBC* by retaining more PIF5 in the enlarged PBs.

**Reducing PB size accelerates PIF5 degradation**

If the formation of PBs phase separates the opposing PHYB signaling actions of PIF5 degradation and stabilization, altering PB size and, therefore, the PB/nucleoplasm partitioning of PHYB and PIF5 should be able to shift the balance of the two antagonistic PHYB actions and result in a change in the level of PIF5 and its signaling output. For example, if the above model is correct, reducing PB in *mCherry-PIF5/PBC* under dimmer light should increase the nucleoplasmic fractions of PHYB and PIF5 and rebalance PHYB signaling from stabilizing

PIF5 to degrading PIF5. To test this hypothesis, we compared the size and dynamics of PBs and PIF5 degradation in *mCherry-PIF5/PBC* #9 grown under 10 μmol m⁻² s⁻¹ or 0.5 μmol m⁻² s⁻¹ R light. As expected, the dimmer R light reduced PHYB activity and promoted hypocotyl elongation in Col-0 (Fig. 5a, b). The *mCherry-PIF5/PBC* seedlings grown under 0.5 μmol m⁻² s⁻¹ R light were also significantly taller than the *mCherry-PIF5/PBC* seedlings grown under 10 μmol m⁻² s⁻¹ and *PBC* seedlings grown 0.5 μmol m⁻² s⁻¹ R light (Fig. 5a, b), which suggests that mCherry-PIF5 in *mCherry-PIF5/PBC* was more abundant and/or more active under the dim light. As expected, PHYB-CFP PBs became significantly smaller under the dim light (Fig. 5c, d)[28]. Interestingly, mCherry-PIF5 was detectable using confocal microscopy in only 75.3% of PBs in the dim light (Fig. 5e) as opposed to in 100% PBs under strong R light (Fig. 4e), suggesting that mCherry-PIF5 was less stable in dim light. Supporting this conclusion, the steady-state levels of both mCherry-PIF5 and endogenous PIF5 decreased dramatically in *mCherry-PIF5/PBC* under 0.5 μmol m⁻² s⁻¹ R light compared to 10 μmol m⁻² s⁻¹ R light (Fig. 5f). The degradation kinetics of mCherry-PIF5 were dramatically increased in *mCherry-PIF5/PBC* under dim R light compared to that under strong R light (Fig. 5g). PHYB-CFP became more dynamic in the small PBs under 0.5 μmol m⁻² s⁻¹ R light, suggesting that dimmer light promoted a gel-to-liquid transition (Fig. 5h). Intriguingly, the dynamics of mCherry-PIF5 stayed the same between high and low light conditions (Fig. 5i). These results suggest that although the large and small PBs might have different PIF5-binding capacities, their PIF5 binding affinities were similar. Therefore, reducing the PB size promoted PIF5 degradation likely by enhancing

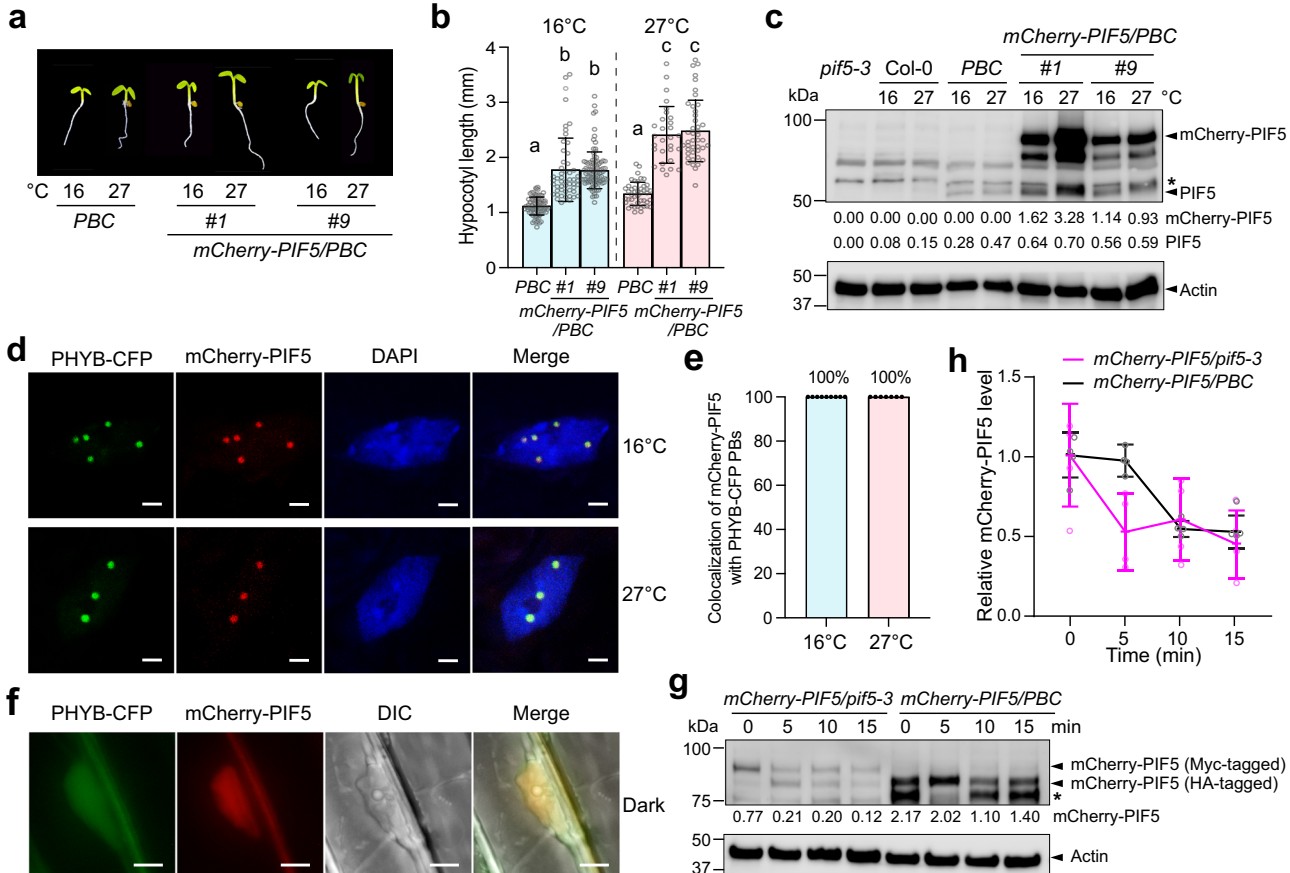

**Fig. 4 | PHYB recruits and stabilizes PIF5 in PBs. a** Images of 4-d-old *PBC* and *mCherry-PIF5/PBC* (#1 and #9) seedlings grown under 10 μmol m⁻² s⁻¹ R light at either 16 °C or 27 °C. **b** Hypocotyl length measurements of the seedings described in (**a**). Error bars represent the s.d., and the centers represent the mean (n ≥ 32 seedlings). Samples labeled with letters exhibited statistically significant differences in hypocotyl length (ANOVA, Tukey's HSD, multiplicity adjusted *p* < 0.05). **c** Immunoblots showing the steady-state levels of PIF5 and mCherry-PIF5 in 4-d-old *pif5-3*, Col-0, *PBC*, and *mCherry-PIF5/PBC* (#1, #9) seedlings grown under 10 μmol m⁻² s⁻¹ R light at either 16 °C or 27 °C. **d** Confocal images showing the colocalization of mCherry-PIF5 with PHYB-CFP in PBs in cotyledon pavement cells from 4-d-old *mCherry-PIF5/PBC #9* seedlings grown under 10 μmol m⁻² s⁻¹ R light at either 16 °C or 27 °C. Scale bars are equal to 2 μm. **e** Quantification of the colocalization of mCherry-PIF5 foci with PHYB-CFP PBs in the experiments described in (**d**). Error bars represent the s.e. (*n* = 9 biological replicates for the 16 °C samples, *n* = 7 biological replicates for the 27 °C samples), and the centers represent the mean.

**f** Fluorescence microscope images showing the localization of PHYB-CFP and mCherry-PIF5 in hypocotyl epidermal cells from 4-d-old dark-grown *mCherry-PIF5/PBC #9* seedings. The DIC image shows the nucleus. Scale bars are equal to 5 μm. **g** Immunoblots showing the degradation kinetics of Myc-tagged mCherry-PIF5 in *mCherry-PIF5/pif5-3* and HA-tagged mCherry-PIF5 in *mCherry-PIF5/PBC*. 4-d-old seedlings of *mCherry-PIF5/pif5-3 #8* and *mCherry-PIF5/PBC #9* were incubated with cycloheximide and collected at the indicated time points. For (**c**, **g**), PIF5 and mCherry-PIF5 were detected using anti-PIF5 antibodies. Actin was used as a loading control. The asterisk indicates a nonspecific band. The relative levels of PIF5 and mCherry-PIF5 normalized to the corresponding actin levels are shown below each lane. **h** Quantification of the relative mCherry-PIF5 levels shown in (**g**). Error bars represent the s.d. of four independent replicates. The source data underlying the hypocotyl measurements in (**b**), the immunoblots in (**c**, **g**, **h**), and the quantification of the colocalization of mCherry-PIF5 and PHYB-CFP in **e** are provided in the Source Data file.

the partitioning of PHYB and PIF5 to the nucleoplasm compartment. Together, these results provide further evidence supporting our conclusion that recruiting PIF5 to PBs stabilizes PIF5 by preventing its degradation in the surrounding nucleoplasm.

## Discussion

Reports published more than 20 years ago described how light triggers the localization of photoactivated PHYB to discrete PBs, creating two spatially separated pools of PHYB – a concentrated pool in PBs and a dilute pool in the surrounding nucleoplasm – and that PB size depends on the amount of active PHYB and could be directly regulated by light intensity and quality (i.e., the R/FR ratio)[22,23,28]. However, the functional significance of this spatial segregation of active PHYB remained frustratingly elusive. Here, we show that PHYB recruits PIF5 as a client to PBs and, surprisingly, that PHYB exerts two opposing functions in degrading and stabilizing PIF5. Unlike previous studies on PBs, we perturbed PBs by increasing PHYB abundance. This approach allowed us to uncouple the function of PHYB in PBs from that in the

surrounding nucleoplasm. Our results support the model that the condensation of PHYB phase separates the opposing PHYB signaling actions of PIF5 degradation and stabilization into two subnuclear compartments: a PIF5-stabilizing environment in PBs and a PIF5-degrading environment in the surrounding nucleoplasm (Fig. 6a). As such, PB dynamics may regulate the PB/nucleoplasmic partitioning of PHYB and PIF5 and thus enable an environmentally sensitive counterbalancing mechanism for titrating the nucleoplasmic concentration of PIF5 and PHYB signaling outputs (Fig. 6b). Because control of the stability of PHYB-associated signaling components is a major mechanism of light signaling[9,18,40], this PB-dependent counterbalancing mechanism for PIF5 regulation provides a framework for assessing the function of PBs in regulating other PHYB-interacting signaling molecules in light and temperature signaling. We propose that this PB function represents a general function of biomolecular condensates that allows distinct variations of a cellular process or signaling pathway to coexist and interact to generate dynamically adjustable integrated outputs within a single subcellular space.

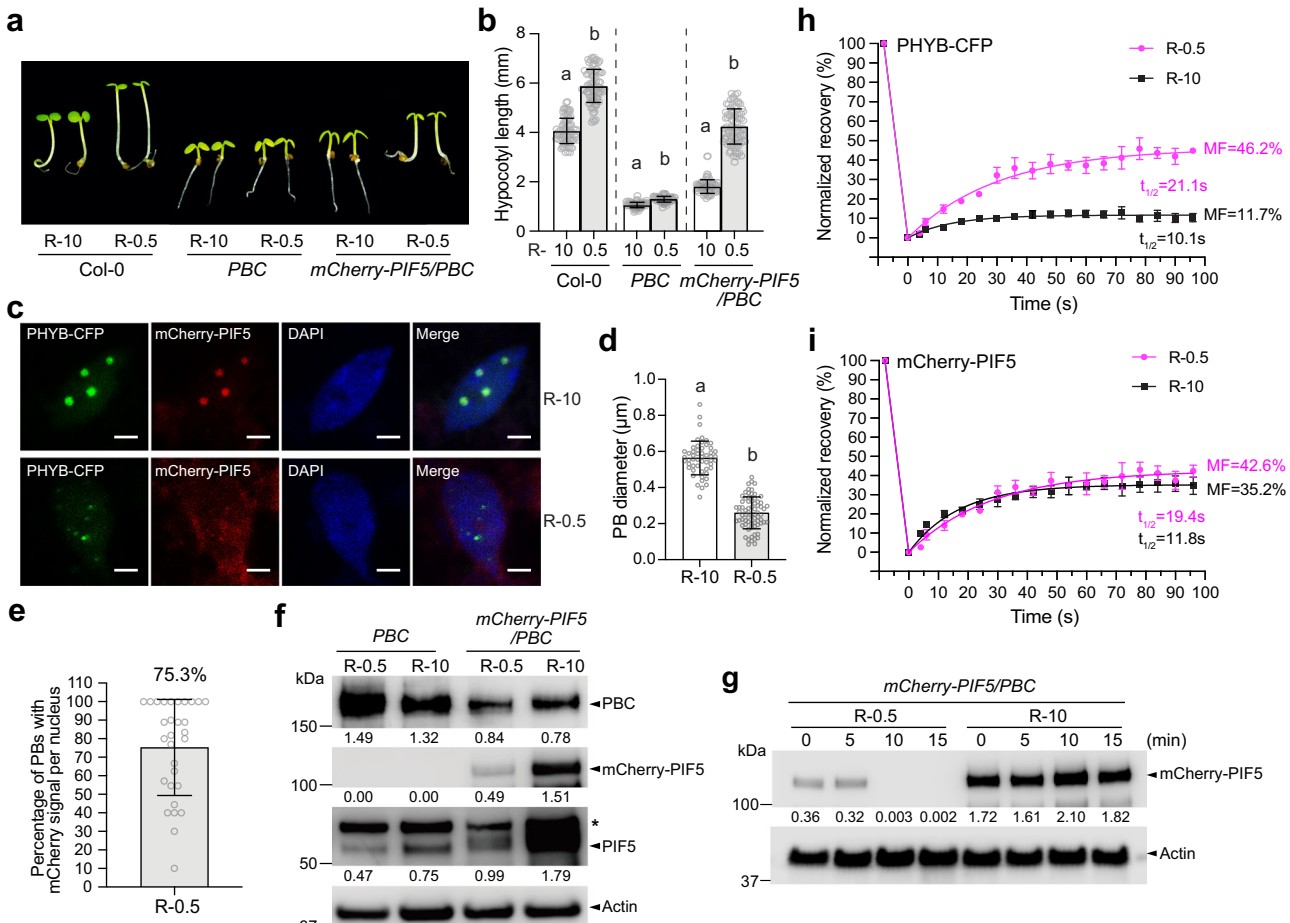

**Fig. 5 | Reducing PB size accelerates PIF5 degradation. a** Images of 4-d-old Col-0, *PBC* and *mCherry-PIF5/PBC #9* seedlings grown under either 10 µmol m$^{-2}$ s$^{-1}$ or 0.5 µmol m$^{-2}$ s$^{-1}$ R light (R-10 and R-0.5, respectively). **b** Hypocotyl length measurement of the seedlings described in (**a**). Error bars represent the s.d., and the centers represent the mean ($n = 90$ seedlings). Samples labeled with different letters exhibited statistically significant differences in hypocotyl length (two-tailed Student's t-test, $p < 0.05$). **c** Confocal images showing the PB patterns in cotyledon pavement epidermal cells from the *mCherry-PIF5/PBC* seedlings described in (**a**). Scale bars equal 2 µm. **d** Quantification of the sizes of the PBs described in (**c**). Error bars represent the s.d., and the center represents the mean ($n = 53$ PBs for the R-10 samples, $n = 72$ for the R-0.5 samples). Different letters denote statistically significant differences in PB size (two-tailed Student's t-test, $p < 0.05$). **e** Quantification of the percentage PHYB-CFP PBs with detectable mCherry-PIF5 fluorescence in *mCherry-PIF5/PBC* seedlings grown under 0.5 µmol m$^{-2}$ s$^{-1}$ R light. The error bar represents the s.d., and the center represents the mean ($n = 30$). **f** Immunoblots of PHYB-CFP, endogenous PIF5 and mCherry-PIF5 in 4-d-old *PBC* and *mCherry-PIF5/PBC #9* seedlings grown under either 10 µmol m$^{-2}$ s$^{-1}$ or 0.5 µmol m$^{-2}$ s$^{-1}$ R light. PHYB-CFP was detected using anti-PHYB antibodies, and mCherry-PIF5 and PIF5 were detected using anti-PIF5 antibodies. Actin was used as a control. The relative levels of PHYB-CFP, endogenous PIF5, and mCherry-PIF5, normalized to the corresponding actin levels, are shown under each lane. The asterisk indicates a non-specific band. **g** Immunoblots showing the degradation kinetics of mCherry-PIF5 in 4-d-old *mCherry-PIF5/PBC #9* seedlings grown under either 10 µmol m$^{-2}$ s$^{-1}$ or 0.5 µmol m$^{-2}$ s$^{-1}$ R light. Seedlings were incubated with cycloheximide and collected at the indicated time points. Total proteins were extracted and subjected to immunoblotting analysis using anti-PIF5 antibodies. Actin was used as a loading control. The relative mCherry-PIF5 levels normalized to the corresponding levels of actin are shown. **h** Results of FRAP experiments showing normalized fluorescence recovery plotted over time for PHYB-CFP in the *mCherry-PIF5/PBC #9* seedlings described in (**a**). **i** Results of FRAP experiments showing normalized fluorescence recovery plotted over time for mCherry-PIF5 in the *mCherry-PIF5/PBC #9* seedlings described in (**a**). For (**h**, **i**), values represent the mean, and error bars represent the s.e. of at least three biological replicates. Solid lines represent the exponential fit of the data. MF: mobile fraction. $t_{1/2}$ represents the half-time of fluorescence recovery. The source data underlying the hypocotyl measurements in (**b**), the PB characterization in (**d**, **e**), the immunoblots in (**f**, **g**), and the FRAP results in (**h**, **i**) are provided in the Source Data file.

Our results indicate that PHYB recruits PIF5 to PBs for PIF5 stabilization. We demonstrated for the first time that PIF5 localizes to PBs. This conclusion is supported by the colocalization of mCherry-PIF5 with endogenous PHYB in PBs in *mCherry-PIF5/pif5-3* and with PHYB-CFP PBs in *mCherry-PIF5/PBC* (Figs. 2d, e, 4d, e). We showed that the slow maturation time of FPs poses a major obstacle in observing FP-tagged PIFs in live cells. The short-lived mCherry-PIF5 in the *mCherry-PIF5/pif5-3* lines could only be detected using immunofluorescence staining, not via confocal microscopy (Fig. 2d). This technical hurdle might explain the surprisingly low number of detailed investigations into the subcellular localization of FP-tagged PIFs compared with the overwhelming number of reports on PIFs' functions. Although the slow

maturation of mCherry hindered our ability to observe mCherry-PIF5, we serendipitously found that the detectability of mCherry fluorescence could be used as an internal reporter to assess the stability or half-life of mCherry-PIF5 in live cells. When PIF5 degradation was attenuated in *mCherry-PIF5/PBC* (Fig. 4g), the fluorescence of mCherry-PIF5 became detectable using confocal microscopy (Fig. 4d). The fact that longer-lived mCherry-PIF5 in *mCherry-PIF5/PBC* was colocalized with PHYB-CFP in PBs provides direct evidence demonstrating that mCherry-PIF5 was retained and stabilized in PBs (Fig. 4d, e). Interestingly, mCherry-PIF5 in *mCherry-PIF5/PBC* localized to PBs only in the light but not in darkness (Fig. 4d, f), supporting the idea that PIF5 was recruited to PBs by PHYB, likely via direct interaction[45]. This conclusion

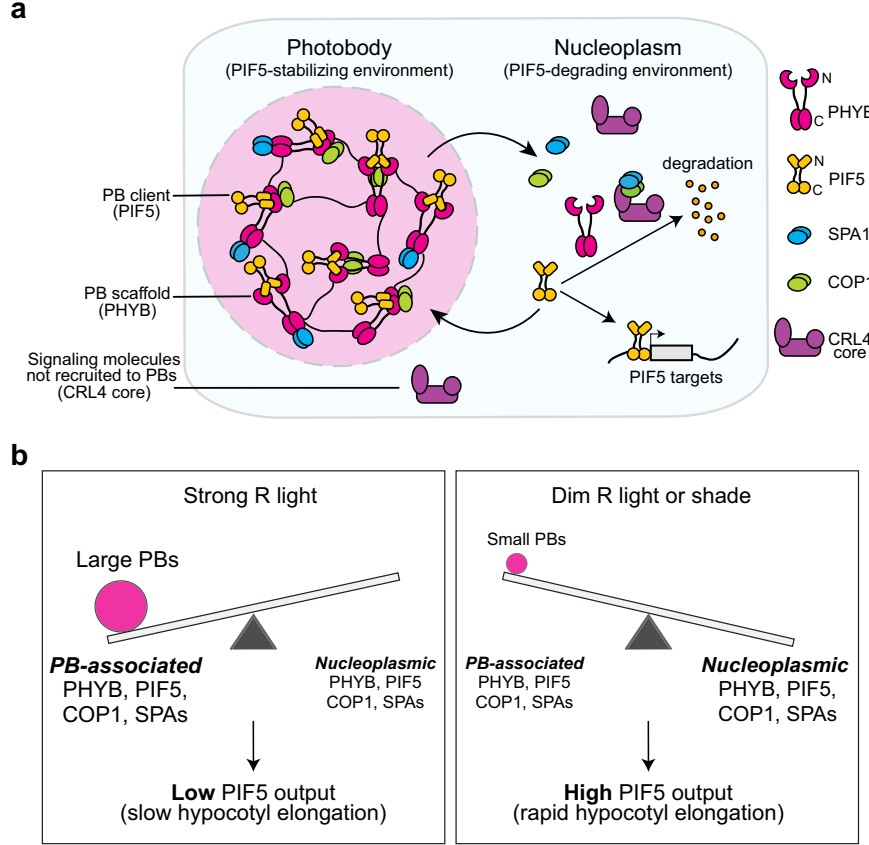

**Fig. 6 | A PB-enabled counterbalancing model for titrating PHYB-mediated environmental responses. a** Schematic illustration of the function of PBs in segregating the opposing PHYB signaling actions of PIF5 degradation and stabilization. The condensation of PHYB creates a PIF5-stabilizing environment in PBs to counteract PIF5 degradation in the surrounding nucleoplasm. PHYB promotes PIF5 degradation in the nucleoplasm by triggering its phosphorylation and subsequent ubiquitylation by the CRL4[COP1/SPA] E3 ubiquitin ligases[45,46,48]. PHYB serves as a scaffold component to recruit PIF5 as a client to PBs via direct interaction. PHYB stabilizes PIF5 by selectively recruiting COP1 and SPAs, but not the CRL4 core, to PBs[21,26,71]. The high concentration of PHYB in PBs disrupts the COP1-SPA complex via distinct interactions of COP1 with PHYB's N-terminal photosensory module and SPAs with PHYB's C-terminal output module[52–54,68,70]. **b** Model for a PB-enabled counterbalancing mechanism to titrate nucleoplasmic PIF5 and environmental responses. Changes in the intensity and composition of light (and also in temperature) directly control the amount of active PHYB and thus the size of PBs to regulate the PB-to-nucleoplasm partitioning of PHYB, PIF5, and other PB constituents, thereby titrating the nucleoplasmic PIF5 and its signaling output. Strong R light increases the amount of active PHYB and enlarges PBs, thereby sequestering a greater fraction of PHYB and PIF5 in PBs, which stabilizes PIF5 and simultaneously reduces the nucleoplasmic PIF5 and its signaling output. This mechanism allows hypocotyl to grow slowly in the light. Dim R light or shade reduces the amount of active PHYB and PB size, thereby enhancing the nucleoplasmic fraction of PHYB and PIF5, simultaneously promoting PIF5 degradation and its transcriptional output. The latter mechanism accelerates hypocotyl elongation.

corroborates the proteomics results that PBs comprise PHYB and its primary and secondary interacting signaling components[21,26]. PIF5 was not identified in the reported proteomics analysis of PB components, which may be due to the timing and conditions used in those studies, as the expression of PIF5 was controlled by the circadian clock[66]. One concept of biomolecular condensates is the concept of scaffolds and clients[67]. Scaffold molecules are considered the drivers of phase separation, whereas molecules recruited into biomolecular condensates formed by scaffolds are called clients[67]. The current data suggest that PHYB is a scaffold component, whereas PIF5 is a client component recruited to PBs, likely via direct interactions with PHYB and other PB components (Fig. 6a). Supporting this model, altering PB size by manipulating the concentration of active PHYB changed the dynamics of the scaffold PHYB but not that of PIF5 (Fig. 5h, i), suggesting that changing PB size altered only PB's binding capacity but not affinity to PIF5. Together, our results reveal that in addition to promoting PIF5 degradation, PHYB stabilizes PIF5 by recruiting PIF5 to PBs, providing a mechanism that allows PIF5 to accumulate in the light.

PHYB-mediated degradation of PIFs has been invoked as one the most important mechanisms of light signaling[18,40]. PIF5 is relatively stable in dark-grown seedlings, and upon exposure to light, PHYA and PHYB induce the rapid phosphorylation and degradation of PIF5[45,46,48]. However, the site of PIF5 degradation has yet to be previously studied. Our results support the model that the PHYB condensation phase separates the PIF5-stabilizing environment in PBs from the PIF5-degrading environment in the surrounding nucleoplasm (Fig. 6a). The mechanism that distinguishes the PIF5-stabilizing environment in PBs and the PIF5-degrading environment in the surrounding nucleoplasm requires further investigation. Previous studies have suggested that PIF5 degradation is mediated by the CRL4[COP1/SPA] E3 ubiquitin ligase[48]. PHYB interacts directly with COP1 and SPAs via different domains: while the N-terminal photosensory module interacts with COP1[68], the C-terminal output module confers a strong interaction with SPAs[52,53,69]. As such, PHYB inhibits the activity of CRL4[COP1/SPA] by blocking the COP1-SPA interaction[52–54,68,70]. Both COP1 and SPAs have been identified as PB components[21,26]. It is likely that the high concentration of PHYB in PBs facilitates the dissociation of COP1-SPA complexes (Fig. 6a). In addition, cullin4 was not identified as a PB constituent[26,57] and, instead, was shown to disperse evenly in the nucleoplasm[71]. Therefore, it is possible that the functional CRL4[COP1/SPA] complex could be assembled only in

the nucleoplasm, providing another possible mechanism for the distinct functions between PBs and the surrounding nucleoplasm (Fig. 6a). In this model, the dynamic changes of PBs would also regulate the amount of functional CRL4$^{COP1/SPA}$ complexes in the nucleoplasm.

Our results suggest that PHYB condensation may enable an environmentally sensitive counterbalancing mechanism to titrate environmental responses (Fig. 6b). Changes in light intensity and composition (and also temperature) directly control the amount of active PHYB and the size of PBs, which could regulate the PB-to-nucleoplasmic partitioning of PHYB, PIF5, and other PB constituents, thereby titrating the nucleoplasmic PIF5 and its signaling output (Fig. 6b). Supporting this model, PB size correlated positively with PIF5 stabilization and negatively with PIF5 degradation. A transition from small PBs in *mCherry-PIF5/pif5-3* to large PBs in *mCherry-PIF5/PBC* switched the balance of PHYB signaling from degrading to stabilizing PIF5 (Figs. 2–4). Conversely, transferring *mCherry-PIF5/PBC* seedlings to dim light, which induced the transition from large PBs to small PBs, accelerated PIF5 degradation (Fig. 5). The current data suggest that localization of PIF5 to PBs impedes its functions in promoting hypocotyl elongation. This model corroborates the proposed function of PBs for sequestering PIF7[72,73]. Enhancing the nucleoplasmic pool of mCherry-PIF5 in dim light promoted hypocotyl elongation (Fig. 5a, b). However, counterintuitively, the levels of both endogenous PIF5 and mCherry-PIF5 in *mCherry-PIF5/PBC* were reduced in the dim light compared to those in the strong light condition (Fig. 5f), indicating that the steady-state level of PIF5 does not correlate with its function in promoting hypocotyl elongation, likely due to enhanced degradation in the nucleoplasm. Another possibility is that PIF5 degradation is associated or coupled with its transcription activation activity. A similar mechanism was proposed for PIF3 as blocking PIF3 degradation in the *hmr* mutant attenuates the activation of its target genes[60].

The function of PBs in stabilizing PIF5 may not be extrapolated to other PIFs, as PIFs are degraded via distinct mechanisms. The degradation of PIF1 is mediated by CRL4$^{COP1/SPA}$ and CRL1$^{CTG10}$ with the F-box protein COLD TEMPERATURE GERMINATION 10 (CTG10) as the substrate recognition subunit[69,74], whereas PIF3 is degraded by CRL3$^{LRB}$ and CRL1$^{EBF}$. PBs were proposed to be required for PIF1 and PIF3 degradation[7,24,29,58,59,61,62]. However, it remains unclear whether PIF3 is degraded in PBs. Although PIF4 can also accumulate in the light[36,46,55], PIF4 is ubiquitylated by the cullin3-based E3 ligase CRL3$^{BOP1/2}$ using BLADE ON PETIOLE (BOP1/2) as the substrate recognition components[75]. When coexpressed with PHYB, FP-tagged PIF4, as well as PIF3 and PIF6/PIL2, localized with PHYB in biomolecular condensates in mammalian cells or protoplasts in a PHYB-dependent manner, supporting the idea that PHYB may be able to recruit PIF4 to PBs[37,38]. However, the PB localization of PIF4 in *Arabidopsis* has not been carefully studied, and the potential role of PBs in stabilizing PIF4 in the light remains to be experimentally verified.

A major difference separating this study from the previous investigations about PBs is the approach used to perturb PBs. Because disrupting PB assembly using loss-of-function mutants in PHYB and PHYB signaling could not uncouple PHYB signaling outputs in PBs and the surrounding nucleoplasm, here we perturbed the PB size by increasing PHYB abundance. If the LLPS of PHYB had driven PB formation, increasing the PHYB level would be expected to only enlarge the PB size without changing the concentrations of PHYB in the nucleoplasm[4,63]. Consistent with the PHYB LLPS model, increasing PHYB abundance enlarged the PB size (Fig. 1d, e). Most PHYB-CFP in PBs was immobile in *gPBC-25* (Fig. 1g). This attribute of PBs was similar to the biomolecular condensates formed by PHYB alone in mammalian cells[37]. Consistently, increasing the PHYB level in *PBC* further reduced the fluorescence recovery kinetics and the mobile fraction of PHYB-CFP in PBs, indicative of a gel-like state (Fig. 1g). These results suggest that changes in PHYB abundance under the physiological PHYB concentration may alter both PB

size and dynamics. Another unexpected result is that increasing PHYB abundance elicited a biphasic PIF5 response, implying that the nucleoplasmic PHYB increased in *gPBC-25* to promote PIF5 degradation. One possibility is that at the physiological PHYB concentration in Col-0, PHYB condensation may form under the critical concentration via a complex mechanism beyond LLPS; in this scenario, like what was observed, overexpressing PHYB above the physiological PHYB concentration would alter both PB size and the nucleoplasmic PHYB concentration. Although we could not have revealed the function of PBs in stabilizing PIF5 without perturbing PBs using the PHYB overexpressing lines, it is important to note that a limitation of characterizing PB dynamics using PHYB overexpression lines would be the difficulty to completely exclude the possibility of a secondary effect on the signaling output due to PHYB overexpression.

This study reveals a PB-mediated light signaling mechanism in which PB formation spatially segregates two opposing PHYB signaling actions of PIF5 stabilization and degradation, which could enable a counterbalancing mechanism to titrate nucleoplasmic PIF5 and environmental responses (Fig. 6). We propose that this PB function represents a general function of biomolecular condensates to allow distinct variations of a cellular process or signaling pathway to coexist and interact to generate dynamically adjustable integrated outputs within a single subcellular space.

## Methods

### Plant materials, growth conditions, and hypocotyl measurements

The *Arabidopsis* Columbia ecotype (Col-0) was used throughout this study. The *phyB-9*[76], *pifq*[77], and *pif5-3* (SALK_087012)[78] mutants, as well as the *PBC* line[32], were previously described. The *gPBC-25*, *gPBC-29*, *mCherry-PIF5/pif5-3* (#8 and #9) and *mCherry-PIF5/PBC* (#1 and #9) transgenic lines were generated in this study. *Arabidopsis* seeds were surface-sterilized and plated on half-strength Murashige and Skoog (½ MS) medium containing Gamborg's vitamins (MSP06, Caisson Laboratories, Smithfield, UT), 0.5 mM MES pH 5.7, and 0.8% agar (w/v). Seeds were stratified in the dark at 4 °C for five days before treatment with specific light conditions and temperatures in an LED chamber (Percival Scientific, Perry, IA). Seedlings grown in the dark were exposed to 10 µmol m$^{-2}$ s$^{-1}$ FR light for 3 h after stratification to induce germination. Fluence rates of light were measured using an Apogee PS-200 spectroradiometer (Apogee Instruments, Logan, UT) and Spec-traWiz® spectroscopy software (StellarNet, Tampa, FL). Images of representative seedlings were taken using a Leica M165 FC stereo microscope (Leica Microsystems, Deerfield, IL) and processed using Adobe Photoshop CC (Adobe, San Jose, CA). For hypocotyl measurements, seedlings were scanned using an Epson Perfection V700 photo scanner (Epson America, Los Alamitos, CA), and hypocotyl length was measured using NIH ImageJ software (http://rsb.info.nih.gov/nih-image/).

### Plasmid construction and generation of transgenic lines

To generate the *gPBC* lines, *PHYB* genomic DNA containing a *CFP* fused in-frame before the stop codon was cloned into *pJHA212G*[79]. The resulting construct was transformed into *phyB-9*. To generate *mCherry-PIF5/pif5-3* lines, a 2.3 kb genomic DNA fragment, encompassing the *PIF5* promoter region, 5' UTR, and the first two codons, was amplified using PCR and subcloned with 4× *Myc*, *mCherry-2-L* and the *PIF5* coding sequence into *pJHA212G*[79] containing the *rbcS* terminator using Gibson Assembly (New England Biolabs, Ipswich, MA). The resulting construct was transformed into *pif5-3* using Agrobacterium-mediated transformation. To generate *mCherry-PIF5/PBC* lines, a 1.2 kb *UBQ10* promoter was amplified using PCR and subcloned with 3× *HA*, *mCherry-2-L* and the *PIF5* coding sequence into *pJHA212B*[79] containing an *rbcS* terminator using Gibson Assembly. The resulting construct was

transformed into *PBC* Agrobacterium-mediated transformation. The primers used for plasmid construction in this study are listed in Supplementary Table 1.

## Protein extraction and immunoblotting

Total protein was extracted from *Arabidopsis* seedlings grown under the indicated conditions. For assessing the degradation kinetics of PIF5 and mCherry-PIF5, seedlings were vacuum infiltrated for 15 min in PBS containing 200 μM cycloheximide and then collected at the indicated time points. At least three independent degradation kinetics experiments were performed to calculate the average relative PIF5 or mCherry-PIF5 levels at each time point normalized to the PIF5 value at time zero. Plant tissues were ground in liquid nitrogen and resuspended in extraction buffer containing 100 mM Tris-HCl pH 7.5, 100 mM NaCl, 5% SDS, 5 mM EDTA pH 8.0, 20 mM dithiothreitol, 20% glycerol, 142 mM β-mercaptoethanol, 2 mM phenylmethylsulfonyl fluoride, 1× cOmplete™ EDTA-free Protease Inhibitor Cocktail (Sigma-Aldrich, St. Louis, MO), 80 μM MG132, 80 μM MG115, 1% phosphatase inhibitor cocktail 3 (Sigma-Aldrich, St. Louis, MO), 10 mM N-ethylmaleimide (NEM), 2 mM sodium orthovanadate ($Na_3OV_4$), 25 mM β-glycerophosphate disodium salt hydrate, 10 mM NaF, and 0.01% bromophenol blue. Protein extracts were boiled for 10 min and then centrifuged at $16,000 \times g$ for 10 min at room temperature. Protein extracts were separated via 12% SDS-PAGE, transferred to a PVDF membrane, probed with the indicated primary antibodies, and then incubated with HRP-conjugated secondary antibodies. Rabbit anti-PIF5 polyclonal antibody (AS12 2112, Agrisera, Vännäs, Sweden) was used at a 1:1000 dilution. Mouse anti-PHYB monoclonal antibody (a gift from Dr. Akira Nagatani) was used at a 1:2000 dilution. Goat anti-mouse (1706516, Bio-Rad Laboratories, Hercules, CA) and goat anti-rabbit (1706515, Bio-Rad Laboratories, Hercules, CA) secondary antibodies were used at a 1:5000 dilution. Signals were detected via SuperSignal West Dura Extended Duration Chemiluminescent Substrate (Thermo Fisher Scientific, Waltham, MA). Western blots were quantified using ImageJ software (http://rsb.info.nih.gov/nih-image/). The relative level of the protein of interest was calculated by normalizing against the corresponding level of the actin control. The protein degradation kinetics curves were generated using data from at least three biological replicates.

## Immunofluorescence staining

Whole-mount immunofluorescence staining was performed as described previously with the following modifications[72,80]. Seedlings were fixed in 4% (v/v) paraformaldehyde (15710, Electron Microscopy Sciences, Hatfield, PA), dehydrated, and mounted on slides[72,80]. All subsequent steps were performed in a 55 μL SecureSeal chamber (621505, Grace Bio-Labs, Bend, OR). Myc-tagged mCherry-PIF5 was detected using rabbit anti-Myc polyclonal antibodies (2272, Cell Signaling Technology, Danvers, MA) as the primary antibody at 1:100 dilution and donkey anti-rabbit AlexaFluor 555 antibodies (A31572, Thermo Fisher Scientific, Waltham, MA) as the secondary antibody at 1:1000 dilution. PHYB was detected using mouse monoclonal anti-PHYB antibodies (a gift from Akira Nagatani, 1:100 dilution) as the primary antibody and donkey anti-mouse AlexaFluor 488 antibodies (A21202, Thermo Fisher Scientific, Waltham, MA, 1:1000 dilution) as the secondary antibody. Nuclei were counterstained with 3.6 μM 4′,6-diamidino-2-phenylindole (DAPI). Samples were mounted using Pro-Long Gold Antifade Mountant (Thermo Fisher Scientific, Waltham, MA) and left to cure overnight in the dark before confocal analysis.

## Seedling preparation for PB analysis using confocal microscopy

Seedlings were fixed following a previously described protocol with slight modifications[81]. Seedlings were fixed under vacuum with 1% (v/v) paraformaldehyde in PBS for 10 min. After quenching with 50 mM $NH_4Cl$, the fixed seedlings were permeabilized with 0.2% Triton X-100

in PBS, and nuclei were stained with 3.6 μM DAPI in PBS for 10 min. Seedlings were washed with PBS before being mounted on a slide using Prolong Diamond Antifade Mountant (Thermo Fisher Scientific, Waltham, MA). The slides were left to cure overnight in the dark before being sealed with nail polish and stored at 4 °C. For live-cell imaging, seedlings were mounted using PBS. The slides were transported to the microscope in an aluminum foil-wrapped petri dish, and nuclei were imaged within 5 min after mounting.

## Fluorescence microscopy and imaging analysis

For colocalization of mCherry-PIF5 and PHYB in *mCherry-PIF5/pif5-3* using immunofluorescence staining, three-dimensional (3D) image stacks of individual nuclei of cotyledon pavement epidermal cells were imaged using a Zeiss LSM800 confocal microscope equipped with a ×100/1.4 Plan-Apochromat oil-immersion objective (Carl Zeiss AG, Jena, Germany). Alexa 488 was monitored using 488 nm excitation and 490–561 nm bandpass emission. Alexa 555 was monitored using 561 nm excitation and 560–630 nm bandpass emission. DAPI was monitored using 405 nm excitation and 410–470 nm bandpass emission. For colocalization mCherry-PIF5/PBC lines, 3D image stacks of individual nuclei of cotyledon pavement epidermal cells were imaged using a Zeiss LSM800 confocal microscope equipped with a ×100/1.4 Plan-Apochromat oil-immersion objective (Carl Zeiss AG, Jena, Germany). PHYB-CFP was monitored using 405 nm excitation and 410–470 nm bandpass emission. mCherry-PIF5 was monitored using 561 nm excitation and 560–630 nm bandpass emission. DAPI was monitored using 405 nm excitation and 410–470 nm bandpass emission. Deconvolution (nearest neighbor) was performed, and the maximum projection of image stacks (Z-stack interval 0.7 μm) was generated using Zeiss ZEN 2.3 software (Carl Zeiss AG, Jena, Germany). Representative images were exported as TIFF files and processed using Adobe Photoshop CC (Adobe, San Jose, CA). To measure the PB diameter of fixed *mCherry-PIF5/PBC* seedlings, the boundary of the PB peak was defined as the point where the fluorescence intensity was half of the peak value.

For PB analysis in *gPBC-25*, *gPBC-29*, *PBC*, and dark-grown *mCherry-PIF5/PBC*, 3D image stacks of individual nuclei from cotyledon epidermal or upper hypocotyl epidermal cells were imaged using a Zeiss Axio Observer Z1 inverted microscope equipped with a Plan-Apochromat 100x/1.4 oil-immersion objective and an Axiocam mono camera (Carl Zeiss, Jena, Germany). Fluorescence was detected using a broad spectrum X-Cite 120LED Boost high-power LED illumination system (Excelitas Technologies, Waltham, MA) and the following Zeiss filter sets: DAPI, excitation 365 nm, emission 445/50 nm/nm (Zeiss Filter Set 49); CFP, excitation 436/25 nm/nm, emission 480/40 nm/nm (Filter set 47 HE); mCherry, excitation 550/25 nm/nm, emission 605/70 nm/nm (Zeiss Filter set 43 HE), brightfield was used for the DIC image. Image stacks with a Z-step size of 0.24 μm were subjected to iterative classic maximum likelihood estimation deconvolution using Huygens Essential (Scientific Volume Imaging, Hilversum, Netherlands). The volume of PBs was quantified using the Huygens Object Analyzer (Scientific Volume Imaging, Hilversum, Netherlands). The PB partitioning of PHYB-CFP was measured using the Huygens Object Analyzer and 3D image stacks of PHYB-CFP and the DAPI-stained nucleus. Huygens Object Analyzer renders a 3D model of the PHYB-CFP signals in PBs and the entire nucleus that was defined by the DAPI signal. The PB-partitioning of PHYB-CFP was calculated as the percentage of PHYB-CFP in PBs versus the entire nucleus.

For FRAP experiments, seedlings were mounted in PBS and placed onto a Zeiss LSM800 confocal microscope with a C-Apochromat 63×/1.2 W autocorr M27 objective (Carl Zeiss AG, Jena, Germany). For the *gPBC-25* and *PBC* experiments, PHYB-CFP was monitored using 405 nm excitation and 490–561 nm bandpass detector settings. Images were acquired at 2.7× magnification, the image format was 512 × 512 pixels, and the pinhole was set at 97 μm. One image was taken pre-bleach,

followed by 6 iterations of photobleaching using the 405 nm laser at 85% power. Images after bleaching were collected using low laser intensities and were taken at 4 s intervals for 18 cycles. For the *mCherry-PIF5/PBC* experiments, PHYB-CFP was monitored using 405 nm excitation and 490–561 nm bandpass detector settings, and mCherry-PIF5 was excited with a 561 nm laser and observed using a 560–630 nm bandpass emission. Images were acquired at 2.7× zoom, the image format was 512 × 512 pixels, and the pinhole was 121 µm. One image was taken pre-bleach, followed by 8 iterations of photobleaching using the 405 nm laser at 50–60% power for PHYB-CFP and the 561 nm laser for mCherry-PIF5 at 85% power. Images after bleaching were collected using low laser intensities and were taken at 6 s intervals for 20 cycles. FRAP curve analyses were performed by setting the time of bleach to zero. Normalization was performed by taking the sum of the intensity of the ROI, subtracting the background, and setting the initial pre-bleach intensity to 1. Data were fitted to a one-phase exponential curve, and MF and $t_{1/2}$ were calculated using Prism 10 (Graph-Pad Software, Boston, MA).

### RNA extraction and quantitative real-time PCR

Total RNA from seedlings was isolated using a Quick-RNA MiniPrep kit with on-column DNase I treatment (Zymo Research, Irvine, CA). cDNA was synthesized using Oligo(dT) primers and 1 µg total RNA via a Superscript II First-Strand cDNA Synthesis Kit (Thermo Fisher Scientific, Waltham, MA). Quantitative real-time PCR (qRT-PCR) was performed with iQ SYBR Green Supermix (Bio-Rad Laboratories, Hercules, CA) using a LightCycler 96 System (Roche, Basel, Switzerland). The transcript level of each gene was normalized to that of *PP2A* (At1G13320). The statistical analyses were performed using one-way analysis of variance (ANOVA) with post hoc Tukey's HSD using Prism 10 (GraphPad Software, Boston, MA). The primers for qRT-PCR analysis used in this study are listed in Supplementary Table 2.

### Reporting summary

Further information on research design is available in the Nature Portfolio Reporting Summary linked to this article.

## Data availability

*Arabidopsis* mutants and transgenic lines as well as plasmids generated during the current study are available from the corresponding author upon reasonable request. Source data are provided with this paper.

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

## Acknowledgements

We thank Akira Nagatani (Kyoto University) for providing anti-PHYB antibodies. We thank Elise Pasoreck for valuable suggestions and comments on the manuscript. We are grateful to Xuemei Chen (Peking University) for her support of D.F. in a collaborative project on thermomorphogenesis. This work was supported by National Institute of General Medical Sciences grant R01GM087388 to M.C.

## Author contributions

R.J.K., D.F., J.H., K.K., J.D. and M.C. conceived the original research plan; M.C. supervised the experiments; R.J.K., D.F., J.H., K.K., J.D. and M.C. performed the experiments; R.J.K., D.F., J.H., K.K., J.D. and M.C. analyzed the data; R.J.K., D.F., J.H. and M.C. wrote the article with contributions from all authors.

## Competing interests

The authors declare no competing interests.
