## [Peer Review File · Nature Communications]

Photobody formation spatially segregates two opposing phytochrome B signaling actions of PIF5 degradation and stabilizationReviewer #1 (Remarks to the Author):

Dr. Meng Chen's group reported in this manuscript phyB biomolecular condensates impose a biphasic regulatory role on PIF5, an important negative regulator of photomorphogenesis in the PIF family. PIF5 protein stability has two different fates: to be degraded in the nucleoplasm under lower phyB (smaller condensates) and to be sequestered and stabilized under higher phyB (larger condensates). The authors claimed that this biphasic regulation might be applicable to the regulation of additional signaling molecules interacting with phyB, together constituting a sophisticated and dynamic regulatory mechanism. In general, the data and the implications are intriguing. The concept put forward here could be game-changing for studying PB-regulated light responses, therefore, it's important to carefully scrutinize the experimental approaches and data interpretation to back up the theory proposed. Below I listed a few points to be clarified before it's considered for publication.

Major points:

1. Most of the data presented are based on phyB (driven by 35S) and PIF5 (under both endogenous and ubiquitin promoters) at levels much higher than their endogenous counterparts. It would be helpful to corroborate if the interesting observation made represents "possibilities of action modes", but not artifacts. For example, measure PIF5 levels under particular circumstances (light/temperature) or in cell types with endogenous PBs of similar sizes as in PBC lines? Alternatively, is it possible to induce the formation of larger PBs under given physiological conditions and measure PIF5 levels? Otherwise, the conclusion drawn might be biologically irrelevant.
2. According to the phytochrome nomenclature (Quail et al 1994), PHYB means apophytochrome B. Holophytochrome B should be designated as phyB throughout the manuscript. This leads to a broader question of whether the 40- to 65-fold phytochrome B produced in the PBC lines is holophytochrome B (with covalently bound chromophores and photoactive) and whether the photobodies (PBs) examined here are comprised of only Pfr phyB in homodimers as claimed?
3. In lines 164-167, "Thus, the overproduction of PHYB is expected to result only in the growth of PBs without changing the concentration of PHYB in either the PB or the surrounding nucleoplasmic compartments and, therefore, may only enhance the signaling output from PBs while leaving the functional output of the nucleoplasmic PHYB unchanged or less affected." Considering the condensates can be relatively more fluidic, gel, or solid types, and the seeding/formation of the condensates is concentration-dependent. It may be presumptuous to assume the phyB concentration in the nucleoplasm and various sizes of PBs are unchanged when perturbing the phyB protein levels in different transgenic lines. For example, phyB might be at a higher concentration in large PBs and in the nucleoplasm in PBCs.
4. This current study often assumed a significant fraction of the exogenously increased phyB will be recruited or form PBs (eg. Lines 209-210). However, only up to 21% of phyB were in PBs (Fig. 1f) in PBC lines. The relative PIF5 in PBs can be similarly measured but this was not performed. This measurement will provide evidence to infer whether PIF5 is degraded exclusively in the nucleoplasm, or the degradation can occur, at least in the small PBs or the peripheries of larger gel-type PBs? With the large PB sizes in PBC lines, it might be possible to examine if PIF5s are sequestered to the center of the larger and solid-type PBs so they are less prone to migrate to the surface of PBs to be degraded? The possible steric hindrance of antibody labeling of mCherry-PIF5 may hinder this approach by immunofluorescence staining used in this study. If the maturation time of mCherry is a problem, perhaps using an alternative FP that's compatible with phyB-CFP would make it possible to monitor the protein via a confocal microscope rather than immunofluorescence staining to circumvent the limitation.
5. The immunofluorescence staining and confocal microscopy are two very different methods with different sensitivities depending on the antibodies and microscopes used. The ability to detect PIF5 with immunofluorescence doesn't mean the inability to detect PIF5 under the confocal microscope is due to the imbalanced maturation of mCherry and the fast degradation rate of PIF5. The half-life of PIF5 is 20 min according to reference 55, but the half-life of mCherry-PIF5 appeared to be much shorter in Fig. 4h?
6. Data interpretation in this manuscript sometimes could be overly simplified or not consider alternative explanations. For example, in lines 289-293. The results could be interpreted by the overproduction of PIF5 overburdens the degradation system. Also, whether nucleoplasm-localization of cullin 4 in previous reports concluded CRL4 core only in the nucleoplasm in Fig. 6, a

key theory in this manuscript can be carefully validated.

Minor points.

1. Biological replicates should be provided for all immunoblots shown to show data reproducibility or used to calculate mean and standard deviation.
2. The current writing is in general easy to follow and grasp the key points. However, there is significant repetition in the introduction, results, and discussion. The writing could be more concise and effective.

Reviewer #2 (Remarks to the Author):

Manuscript Kim et al. reports novel regulatory mechanism of PIF5 recruitment into PBs. The story is very short and goes to detail mechanistical explanation how PIF5 is regulated. Authors provided evidence that PHYB can regulate PIF5 stability in a concentration-dependent manner. If there is too high level of PHYB, it recruits and stabilized PIF in PBs, if the level of PHYB is lower, it causes PIF5 degradation.

The manuscript is interesting, well written and figures are self-explanatory as it should be. The story is rather short, so maybe it could be considered as letter rather than a full research article. I do have few minor points:

Suggested experiment to figure 1- increase in PHYB amount causes PB to be more solid (FRAP signal does not recover well). I would like to see if those PBs are reversible and if PHYB can dissociate from them at the same time the regulation on PIF5 should change.

I also wonder what would happen if authors could modify PHYB to not be present in the PBs, how that would affect PIF5 stability?

Based on results, it seems modulation of PHYB level in the nuclear PBs can determine stability/degradation of PIF5. In which natural conditions such a switch would occur (here there is higher expression of protein level). Second question related to that, is this regulation PIF5 specific? I missed at least one or two other PIFs as control for the mechanism.

Line 41-45, the sentence should be divided in two, its too long. Suggestion to start new sentence from Therefore...

Line 121-124, again I would skip; and start new sentence from This.

Figure 1f. It is not clear to me how PHYB portioning into PB was calculated.

In the structure of the manuscript, I miss the link between first chapter on PHYB level and second chapter on PIF5. Perhaps authors could make the story line better and link those two chapters in a better way. Other suggestion is to add few transitional sentences to make it clear, why authors went for PIF5..

Figure 3b and c – the correlation profile is very interesting.

Figure 3e – it would be great if authors could add in m&m explanation of WB quantification method.

Line 259-261 – figure panel reference should be added at the end of the sentence, not in the next one.

Figure 5 d,b – according to Nature rules, points of single measurement should be included if $n \leq 10$. Please revise if I am correct. This point applies to multiple graphs throughout all figures.

Response to Reviewers

We thank both reviewers for their thorough reviews of the manuscript and for their constructive comments and suggestions. We have made substantial changes to the manuscript based on the reviewer comments. We also revised the model (Fig. 6) to illustrate the function of PB dynamics in regulating environmental responses.

Reviewer #1

Dr. Meng Chen's group reported in this manuscript phyB biomolecular condensates impose a biphasic regulatory role on PIF5, an important negative regulator of photomorphogenesis in the PIF family. PIF5 protein stability has two different fates: to be degraded in the nucleoplasm under lower phyB (smaller condensates) and to be sequestered and stabilized under higher phyB (larger condensates). The authors claimed that this biphasic regulation might be applicable to the regulation of additional signaling molecules interacting with phyB, together constituting a sophisticated and dynamic regulatory mechanism. In general, the data and the implications are intriguing. The concept put forward here could be game-changing for studying PB-regulated light responses, therefore, it's important to carefully scrutinize the experimental approaches and data interpretation to back up the theory proposed. Below I listed a few points to be clarified before it's considered for publication.

Major points:

1. Most of the data presented are based on phyB (driven by 35S) and PIF5 (under both endogenous and ubiquitin promoters) at levels much higher than their endogenous counterparts. It would be helpful to corroborate if the interesting observation made represents "possibilities of action modes", but not artifacts. For example, measure PIF5 levels under particular circumstances (light/temperature) or in cell types with endogenous PBs of similar sizes as in PBC lines? Alternatively, is it possible to induce the formation of larger PBs under given physiological conditions and measure PIF5 levels? Otherwise, the conclusion drawn might be biologically irrelevant.

Response:

The functions of PIF5 in light and temperature signaling as well as its regulation by PHYB have been extensively studied previously. The published results indicate that 1) PHYB promotes rapid PIF5 degradation in the light [Shen et al. (2007) *Plant Physiol* 145(3):1043; Lorrain et al. (2007) *Plant J* 53:312; Pham et al. (2018) *Plant J* 96(2):260] and 2) PIF5 can also accumulate in prolonged light conditions [Lorrain et al. (2007) *Plant J* 53:312; Qiu et al. (2017) *Nat Commun* 8:1905; Pham et al. (2018) *Plant J* 96(2):260; Sharma et al. (2019) *Nat Commun* 10:4417;], though how PIF5 is stabilized in the light was unknown.

The focus of this study is to draw a mechanistic connection between photobodies (PB) and the previously reported PHYB-mediated PIF5 degradation and the mysterious PIF5 stabilization in the light, particularly to address the question of whether PHYB mediates PIF5 degradation in PBs. As stated in the manuscript, based on our more than 20 years of experience in studying PBs, one thing we have learned is that perturbing PBs by altering environmental conditions or mutating PHYB signaling components cannot distinguish the signaling outputs of PHYB in PBs and the surrounding nucleoplasm. One alternative approach would be to perturb PBs by increasing the amount of PHYB. The theory of the liquid-liquid phase separation (LLPS)

model for PHYB condensation would predict that increasing PHYB above the critical concentration for LLPS should only change the PB size without changing the concentration of PHYB in PBs and the surrounding nucleoplasm [Banani et al. (2017) *Nat Rev Mol Cell Biol* 18:285; Klosin et al. (2020) *Science* 367(6476):464]. This study represents the first attempt to use this new approach to investigate the function of PBs or, to the best of our knowledge, any nuclear bodies in plants. Our results reveal the function of PBs in the physiological subnuclear condition, because:

1. We showed that PHYB forms PBs under the physiological concentration using immunostaining (Fig. 2D) [Willige et al. (2021) *Nat Genet* 53:955].
2. We demonstrated that PIF5 localizes to PBs when PHYB is in the physiological concentration (Fig. 2D).
3. We showed that perturbing PHYB abundance and PB size evokes a biphasic response of the endogenous PIF5 (Fig. 3). These results reveal that PHYB exerts two opposing roles in both degrading and stabilizing endogenous PIF5. The latter provides a new mechanism for the previously described PIF5 accumulation under prolonged light conditions and also leads to the new hypothesis that PHYB stabilizes PIF5 in PBs.
4. We found unexpectedly that the mCherry fluorescence could be used as a reporter for the stability of mCherry-PIF5 *in vivo*. Using both *in vivo* microscopy of mCherry fluorescence and biochemical assays, we demonstrated that PIF5 is being stabilized in PBs (Fig. 4).
5. Furthermore, we showed that reducing PB size – which was expected to increase the nucleoplasmic fraction of PHYB and PIF5 – promoted PIF5 degradation (Fig. 5), strongly supporting the model that PHYB promotes PIF5 degradation in the nucleoplasm.

Together, these creative approaches allowed us, for the first time, to dissect the PHYB functions between the PBs and the surrounding nucleoplasm, therefore providing a mechanistic explanation for the well-characterized phenomena of PHYB-mediated PIF5 degradation and the mysterious PIF5 stabilization under the physiological conditions in the light.

2. According to the phytochrome nomenclature (Quail et al 1994), PHYB means apophytochrome B. Holophytochrome B should be designated as *phyB* throughout the manuscript. This leads to a broader question of whether the 40- to 65-fold phytochrome B produced in the PBC lines is holophytochrome B (with covalently bound chromophores and photoactive) and whether the photobodies (PBs) examined here are comprised of only Pfr *phyB* in homodimers as claimed?

Response:

We thank the reviewer for the kind reminder. We are aware of this nomenclature convention in the phytochrome field. In fact, we faithfully followed this convention when we published our papers in plant-specific journals [Van Buskirk et al. (2012) *Plant Physiol* 158:52; Van Buskirk et al. (2014) *Plant Physiol* 165:595; Qiu et al. (2015) *Plant Cell* 27:1409; Nevarez et al. (2017) *Plant Physiol* 173:1953; Quint et al. (2023) *Trends Plant Sci* 28:1098]. But, when publishing in general journals, like *Nat Commun*, to accommodate a general audience, we prefer to follow the

widely accepted capitalization convention of gene symbols for eukaryotes defined by the International Protein Nomenclature Guidelines (https://www.ncbi.nlm.nih.gov/genome/doc/internatprot_nomenguide/). Here, “PHYB” refers to the holoprotein. It has been well established that the PHYB apoprotein in the phytochromobilin biosynthesis mutant *hy1hy2* localizes to the cytoplasm in the light, indicating that both nuclear and PB localization of PHYB requires the chromophore [Oka et al. (2011) *Plant Cell Physiol* 52(12):2088]. Therefore, the PHYB molecules in PBs are expected to be the holophytochrome. It is also well established that individual PHYB molecules are constantly being converted between the Pr and Pfr forms and even in monochromatic red light PHYB can not stay in the Pfr state 100% of the time due to its fast thermal reversion [Klose et al. (2020) *Mol Plant* 13:386]. As stated in the manuscript, we can only say that PB formation is driven by the Pfr form of PHYB, the size of PBs correlates with the percentage of PHYB in the Pfr form [Chen et al. (2003) *PNAS* 100(24):14493]. Nobody can claim there is only Pfr PHYB in PBs – because no PHYB molecule stays at the Pfr form 100% of the time.

3. In lines 164-167, “Thus, the overproduction of PHYB is expected to result only in the growth of PBs without changing the concentration of PHYB in either the PB or the surrounding nucleoplasmic compartments and, therefore, may only enhance the signaling output from PBs while leaving the functional output of the nucleoplasmic PHYB unchanged or less affected.” Considering the condensates can be relatively more fluidic, gel, or solid types, and the seeding/formation of the condensates is concentration-dependent. It may be presumptuous to assume the phyB concentration in the nucleoplasm and various sizes of PBs are unchanged when perturbing the phyB protein levels in different transgenic lines. For example, phyB might be at a higher concentration in large PBs and in the nucleoplasm in PBCs.

Response:

This statement is actually not our assumption but a prediction based on the theory of the LLPS model, which was suggested to serve as a mechanism to maintain the concentrations of the phase-separated components in both compartments. [Banani et al. (2017) *Nat Rev Mol Cell Biol* 18:285; Klosin et al. (2020) *Science* 367(6476):464]. Applying the LLPS theory to PBs, if PBs are formed via the LLPS of PHYB, it would predict that increasing PHYB above the critical concentration required for LLPS should only change the PB size without changing the concentration of PHYB in PBs and the surrounding nucleoplasm. Based on the theoretical framework, our experiments also serve as a means to test the LLPS model. As the reviewer pointed out, our results indeed suggest that PB formation *in vivo* requires more complex mechanisms beyond simple LLPS. The biphasic PIF5 response to increases in PHYB abundance implies that the nucleoplasmic concentration of PHYB might have changed when altering the expression of PHYB. Furthermore, our companion study on the spatial positioning of PBs indicates that PB formation in the physiological subnuclear environment requires a nucleation step beyond the random LLPS of PHYB. We revised the beginning paragraph of the Results section and the second paragraph from the end of the Discussion to clarify these points.

4. This current study often assumed a significant fraction of the exogenously increased *phyB* will be recruited or form PBs (eg. Lines 209-210). However, only up to 21% of *phyB* were in PBs (Fig. 1f) in PBC lines. The relative PIF5 in PBs can be similarly measured but this was not performed. This measurement will provide evidence to infer whether PIF5 is degraded exclusively in the nucleoplasm, or the degradation can occur, at least in the small PBs or the peripheries of larger gel-type PBs? With the large PB sizes in PBC lines, it might be possible to examine if PIF5s are sequestered to the center of the larger and solid-type PBs so they are less prone to migrate to the surface of PBs to be degraded? The possible steric hindrance of antibody labeling of mCherry-PIF5 may hinder this approach by immunofluorescence staining used in this study. If the maturation time of mCherry is a problem, perhaps using an alternative FP that's compatible with *phyB*-CFP would make it possible to monitor the protein via a confocal microscope rather than immunofluorescence staining to circumvent the limitation.

Response:

The goal of this study, which is stated at the beginning of the manuscript, is to distinguish the functions of PHYB in PBs and the surrounding nucleoplasm. Based on our knowledge, this is the first time that a study of PBs focuses on the function of PHYB in both PBs and the surrounding nucleoplasm. That was the reason why we measured the PB and nucleoplasmic fractions of PHYB (Fig. 1). We agree with the reviewer that it would be informative if we could also measure the PB and nucleoplasmic fraction of PIF5. However, accurately quantifying the PIF5 level using microscopic approaches is challenging if possible at all. As the reviewer indicated, the level of PIF5 could not be accurately assessed by the mCherry fluorescence due to its slow maturation. In fact, the reason why we used mCherry2-L is because it has the fastest maturation time among the red fluorescent proteins [Balleza et al. (2017) *Nat Methods* 15:47]. We chose to use a red-colored FP because all our PHYB-related constructs were labeled using either CFP or YFP. Now we have learned from this study that the maturation time of FPs represents a major obstacle in observing PIFs, it would be a great idea to use other FPs, with a shorter maturation time, to tag PIFs in future investigations. Immunostaining works well for qualitatively assessing the presence of PIF5 in PBs but cannot be used to quantitatively measure the level of PIF5 due to large variations in background staining between experiments.

5. The immunofluorescence staining and confocal microscopy are two very different methods with different sensitivities depending on the antibodies and microscopes used. The ability to detect PIF5 with immunofluorescence doesn't mean the inability to detect PIF5 under the confocal microscope is due to the imbalanced maturation of mCherry and the fast degradation rate of PIF5. The half-life of PIF5 is 20 min according to reference 55, but the half-life of mCherry-PIF5 appeared to be much shorter in Fig. 4h?

Response:

We agree with the reviewer that, as we pointed out above, immunostaining and confocal are very different approaches. As we have shown, immunostaining could detect short-lived mCherry-PIF5 proteins that were undetectable using confocal microscopy (Fig. 2d). The conclusion that the inability to detect mCherry-PIF5 in *mCherry-PIF5/pif5-3* was drawn not only based on this observation alone. We also assessed the stability of mCherry-PIF5 and PIF5 in *gPBC-25* and

PBC (Fig. 4g) using a biochemical method. The biochemical data corroborated the results on mCherry-PIF5 detectability using confocal microscopy. Together, these results support the conclusion that the inability to detect mCherry-PIF in *pi5-3* background was due to a faster degradation rate compared to mCherry's maturation time.

In the study by Pham et al. [*Plant J* 96(2):260 (2018)], the 20-minute half-life of PIF5 (Figure 1a,c) was measured in Col-0. The published PIF5 degradation kinetics was quite similar to our data on PIF5 degradation in Col-0 as shown in Fig. 3d,e. Fig. 4h shows the degradation kinetics of mCherry-PIF5, which was faster than the endogenous PIF5. This discrepancy could be due to the excess amount of mCherry-PIF5 in the transgenic lines. We added the following sentence in the Results section: "It is important to note that the degradation kinetics of mCherry-PIF5 (Fig. 4g,h) was considerably faster than that of endogenous PIF5 (Fig. 3d,e)⁴⁹, this could be attributed to either the mCherry tag or the overexpression of mCherry-PIF5 and the resulted change in their PB/nucleoplasm partitioning."

6. *Data interpretation in this manuscript sometimes could be overly simplified or not consider alternative explanations. For example, in lines 289-293. The results could be interpreted by the overproduction of PIF5 overburdens the degradation system. Also, whether nucleoplasm-localization of cullin 4 in previous reports concluded CRL4 core only in the nucleoplasm in Fig. 6, a key theory in this manuscript can be carefully validated.*

Response:

It is unlikely that the accumulation of mCherry-PIF5 in the *mCherry-PIF5/PBC* lines was caused by the overproduction of mCherry-PIF5 that overburdened the degradation system. This is because the endogenous PIF5 also accumulated in *PBC* (Fig. 4c). We proposed the model in Fig. 6a based on 1) the published data on CUL4 localization [Chen et al. (2006) *Plant Cell* 18:1991] and 2) the fact that only COP1 and SPAs, but not CUL4, were identified in isolated PBs using proteomics approaches [Huang et al. (2016) *Elife* 5:e13292; Kim et al. (2023) *Nat Commun* 14:1708]. We agree with the reviewer that this hypothesis should be further tested in future investigations.

Minor points.

1. *Biological replicates should be provided for all immunoblots shown to show data reproducibility or used to calculate mean and standard deviation.*

Response:

We included the biological replicates for the immunoblots in the Source Data file.

2. *The current writing is in general easy to follow and grasp the key points. However, there is significant repetition in the introduction, results, and discussion. The writing could be more concise and effective.*

Response:

We thank the reviewer for the advice. We have largely revised the manuscript to reduce repetition.

Reviewer #2:

Manuscript Kim et al. reports novel regulatory mechanism of PIF5 recruitment into PBs. The story is very short and goes to detail mechanistical explanation how PIF5 is regulated. Authors provided evidence that PHYB can regulate PIF5 stability in a concentration-dependent manner. If there is too high level of PHYB, it recruits and stabilized PIF in PBs, if the level of PHYB is lower, it causes PIF5 degradation.

The manuscript is interesting, well written and figures are self-explanatory as it should be. The story is rather short, so maybe it could be considered as letter rather than a full research article.

I do have few minor points:

Suggested experiment to figure 1- increase in PHYB amount causes PB to be more solid (FRAP signal does not recover well). I would like to see if those PBs are reversible and if PHYB can dissociate from them at the same time the regulation on PIF5 should change.

Response:

The fluorescence recovery of PHYB-CFP was low in both gPBC-25 and *PBC*, indicating that only a minor fraction of PHYB-CFP molecules in PBs were mobile – this is a unique feature of PBs. Although PHYB-CFP became less dynamic in *PBC* under strong R light, transferring *PBC* to dim R light did transition large PBs to small PBs as expected (Fig. 5c,d) and resulted in enhanced PHYB-CFP dynamics (Fig. 5h). We also wanted to point out that the *PBC* line used in this study has been characterized in previous studies [Chen et al. (2005) *Curr Biol* 15:637]. As such, all data on *PBC* can be cross-referenced. We have shown that *PHYB-CFP* in *PBC* responds to changes in light and temperature similarly to another well-characterized PHYB-GFP line (*PBG*) [Hahm et al. (2020) *Nat Commun* 11:1660].

I also wonder what would happen if authors could modify PHYB to not be present in the PBs, how that would affect PIF5 stability?

Response:

This is a very interesting and complicated question. Many *phyB* mutants have been identified to be defective in PB localization [Chen et al. (2003) *PNAS* 100:14493; Zhang et al. (2013) *Plant Physiol* 161:1445; Qiu et al. (2017) *Nat Commun* 8:1905; Viczian et al. (2020) *New Phytol* 225:1635]. The problem with these mutants is that they were all loss-of-function alleles and were expected to lose the function of PHYB in the nucleoplasm. So far, there is no known *phyB* mutant that is defective in PB formation only while leaving the PHYB function intact. Such a *phyB* mutant is expected to reduce the steady-state level of at least PIF5, though it would be difficult to predict its hypocotyl phenotype as there is no apparent correlation between the steady-state level of PIFs and hypocotyl growth and enhanced PIF degradation could also increase its transactivation activity. Because currently there is no such a *phyB* mutant, we tried to reduce the PB/nucleoplasm partitioning of PHYB using dim light (Fig. 5). Our results support

the model that enhancing the nucleoplasmic fraction of PHYB and PIF5 promoted PIF5 degradation and reduced the steady-state level of PIF5.

Based on results, it seems modulation of PHYB level in the nuclear PBs can determine stability/degradation of PIF5. In which natural conditions such a switch would occur (here there is higher expression of protein level). Second question related to that, is this regulation PIF5 specific? I missed at least one or two other PIFs as control for the mechanism.

Response:

Please see the responses to the first question of Reviewer #1. The published results indicate that 1) PHYB promotes rapid PIF5 degradation in the light [Shen et al. (2007) *Plant Physiol* 145(3):1043; Lorrain et al. (2007) *Plant J* 53:312; Pham et al. (2018) *Plant J* 96(2):260] and 2) PIF5 can also accumulate in prolonged light conditions [Lorrain et al. (2007) *Plant J* 53:312; Qiu et al. (2017) *Nat Commun* 8:1905; Pham et al. (2018) *Plant J* 96(2):260; Sharma et al. (2019) *Nat Commun* 10:4417;], though how PIF5 is stabilized in the light was unknown. We showed that PHYB forms PBs under the physiological concentration using immunostaining (Fig. 2D) [Willige et al. (2021) *Nat Genet* 53:955]. We also demonstrated that PIF5 localizes to PBs when PHYB is in the physiological concentration (Fig. 2D). In natural conditions, changing R light intensity or altering the R/FR ratio would change the concentration of the Pfr/Pfr of PHYB and, thereby, is expected to change the PB/nucleoplasmic partitioning of PHYB and PIF5. Although we could switch the overall output of PHYB signaling between PIF5-stabilization and PIF5 degradation by manipulating the level of PHYB, these results reflect the function of PBs in balancing PIF5 stabilization and degradation under physiological conditions. We still do not know whether the PB-mediated regulation of PIF5 works also for other PIFs. This is one of the questions that we are trying to answer currently. We discussed this question in the 5th paragraph in the Discussion section.

Line 41-45, the sentence should be divided in two, its too long. Suggestion to start new sentence from Therefore...

Response: We thank the reviewer for the comment. We have revised the sentence.

Line 121-124, again I would skip; and start new sentence from This.

Response: We thank the reviewer for the comment. We have revised the sentence.

Figure 1f. It is not clear to me how PHYB portioning into PB was calculated.

Response: We used Huygens Object Analyzer and 3D image stacks of PHYB-CFP and the DAPI-stained nucleus to quantify PHYB-CFP signals in PBs and the entire nucleus. This allows us to calculate the percentage of PHYB-CFP signals in PBs vs the entire nucleus.

In the structure of the manuscript, I miss the link between first chapter on PHYB level and second chapter on PIF5. Perhaps authors could make the story line better and link those two chapters in a better way.

Response: We thank the reviewer for the suggestion. We added the rationale behind using the PIF5 model in the paragraph.

Other suggestion is to add few transitional sentences to make it clear, why authors went for PIF5..

Response: We added the rationale behind using the PIF5 model in the paragraph.

Figure 3b and c – the correlation profile is very interesting.

Response: We agree that the level of PHYB proteins correlated nicely with the transcript level.

Figure 3e – it would be great if authors could add in m&m explanation of WB quantification method.

Response: We thank the reviewer for the advice. We have added the following in the Methods section: “To measure the degradation kinetics of PIF5 and mCherry-PIF5, seedlings were vacuum infiltrated for 15 min in PBS containing 200 μ M cycloheximide and then collected at the indicated time points. At least three independent degradation kinetics experiments were performed to calculate the average relative PIF5 or mCherry-PIF5 levels at each time point normalized to the PIF5 value at time zero.”

Line 259-261 – figure panel reference should be added at the end of the sentence, not in the next one.

Response: We have revised the position of the reference.

Figure 5 d,b – according to Nature rules, points of single measurement should be included if $n \leq 10$. Please revise if I am correct. This point applies to multiple graphs throughout all figures.

Response: We thank the reviewer for the advice. We have added data points to Figs 3b,c,e and 4h. Fig 5d,b also shows data points. All source data are also included in the Source Data file.

Reviewer #2 (Remarks to the Author):

Dear Authors,

I went through the responses and text again. I keep my impression that the manuscript is novel and well written. Regarding my first comments, it still would be interested to see what happens to PIF5 if PBs are reversed. I appreciate the explanation and support of literature in that matter.

Regarding second comment, authors explained more about PIF5 but did not mention about using as control other PIFs to check for specificity.

I am satisfied with the rest of the comments.

Reviewer #3 (Remarks to the Author):

I have not reviewed the initial version of this manuscript but have been invited to comment on the revised version and the authors response to the reviewers of the first version of the manuscript.

Formation of photobodies is an intriguing feature of the plant photoreceptor phytochrome B (phyB). It has been observed the first time more than two decades ago, but their molecular nature and to a large extent also their physiological function remained elusive. In the work presented in this manuscript, the authors perturbed photobody size by overexpression of phyB. They observed that moderately increased levels of phyB lead to destabilisation of PIF5, an important phyB downstream signalling factor, while strong overexpression of phyB stabilises PIF5. The authors interpretation is that PIF5 is stabilised in phyB photobodies, while it is targeted for degradation when localising to the nucleoplasm.

Overall, this is an interesting manuscript with a very surprising result. The experimental data looks solid and clear, and my concerns are mainly related to the interpretation of the data as detailed in my comments below.

Major comments:

1. Introduction, line 112ff: The authors say that degradation of PIF1/3/4/5 is a central mechanism to control their activity by phyB, but they do not mention that binding of phyB to PIF1 and PIF3 (and potentially other PIFs) inhibits their association with target promoters. It has been shown that NGB (N-terminal half of phyB fused to GUS and an NLS) is active in triggering light signalling, but does not promote the degradation of PIF3. Instead, NGB inhibits association of PIF1/3 with promoters, suggesting that light signalling may not require degradation of PIFs. Thus, it is essential to mention this mechanism and to cite the relevant publication by Giltsu Choil lab: Park et al., 2012, *Plant J.*, 72, 537–546.

2. Introduction, line 170ff: I do not understand this sentence: "PHYB condensation stabilizes PIF5 in PBs to counteract PIF5 degradation in the surrounding nucleoplasm, thereby enabling a counterbalancing mechanism to titrate nuclear PIF5 and its signaling outputs.". What does it mean in simple words?

3. Introduction, line 172ff: As take home message, the authors write "This novel PB-mediated PHYB signaling mechanism provides a framework for regulating a plethora of PHYB-interacting signaling molecules to generate diverse environmental responses in plants.". I don't think that this very broad and general statement is appropriate, given that even PIFs differ regarding accumulation in light, factors required for degradation, etc. It is also interesting that a previous publication by the authors lab (Fig. 1d in Qiu et al., 2017, *Nat. Commun.*, 8, 1905) has shown that levels of PIF3 reversely correlate with levels of phyB, suggesting that the effect observed for PIF5 may even not be common among PIFs. Furthermore, the effect observed for PIF5 occurs in lines that have phyB levels much higher than in WT plants, i.e. the regulatory mechanism suggested by the authors may exist ... but it may not be relevant under natural conditions.

4. Results, line 206ff: The authors say that phyB levels in photobodies increased with overall phyB levels but there are no data on phyB levels in the nucleoplasm. However, for the conclusions of the manuscript, it is prerequisite that levels in the nucleoplasm do not change (or at least do not increase with increasing overall phyB levels). Thus, demonstrating this is essential for the conclusions of the manuscript.

5. Results, p. 268ff: I do not question the data showing that the PIF5 level decreases with a

moderate increase in phyB levels, while it increases with a strong overexpression of phyB. However, I doubt that this effect is relevant under natural conditions. Even in the phyB OX line with 8-fold overexpression, PIF5 levels are reduced (and not increased) compared to wild type, i.e. there would have to be natural conditions under which phyB levels increase >8-fold for the effect to be relevant. What are these conditions?

6. Results, line 280ff: The authors write "These results suggest that PHYB signaling exerts opposing roles in both degrading and stabilizing PIF5, the balance of which can be adjusted by altering the PHYB level."  Please give examples for natural conditions (in wildtype plants) under which phyB levels increase more than 8-fold (8-fold overexpression still promotes PIF5 degradation). If such conditions do not exist, the effect of PIF5 stabilisation at strongly increased phyB levels is physiologically not relevant.

7. Discussion, line 370: "Because control of the stability of PHYB-associated signaling components is a major mechanism of light signaling, the novel PB-dependent counterbalancing mechanism provides the framework for regulating a plethora of PHYB-interacting signaling molecules in diverse plant environmental responses. W"  A) This mechanism only works if phyB is strongly (>8-fold) overexpressed and therefore may be irrelevant under natural conditions (unless the authors can define natural conditions under which phyB levels are upregulated to the required extent in wildtype seedlings/plants). B) The authors have shown the effect for PIF5, not for any other protein. Moreover, from their own publication (Qiu et al., 2017, Nat. Commun., 8, 1905), there is evidence that this effect may not exist for PIF3. Thus, the sentence in line 370ff should be rephrased.

8. Discussion, line 408ff: "Together, our results reveal that in addition to promoting PIF5 degradation, PHYB also stabilizes PIF5 by recruiting PIF5 to PBs, providing a mechanism that allows PIF5 to accumulate in the light."  Please clearly say that this depends on strong overexpression of phyB; even 8-fold overexpression is not sufficient to observe the effect!

9. Discussion, line 411f: "PHYB-mediated degradation of PIFs has been invoked as the central mechanism of light signaling"  I agree that PIF degradation has the potential to affect light signalling. However, Park et al., 2012, Plant J., 72, 537–546 have shown that phyB also controls binding of PIF1 and PIF3 to target promoters and that this mechanism is apparently sufficient to control light signalling in NGB background, where PIF3 is not degraded in light. Furthermore, PIF7 is not primarily regulated by degradation. Thus, the sentence in line 411f is misleading and puts too much emphasis on the importance of degradation, while neglecting regulation of promoter binding. In this context, it is appropriate to mention the work by Park et al., 2012, Plant J., 72, 537–546 and include it as reference.

10. Discussion, line 471ff: "Here, we perturbed the PB size by increasing PHYB abundance. If PB formation is driven by the LLPS of PHYB, increasing the PHYB level is expected to only enlarge the PB size without changing the concentrations of PHYB in the nucleoplasm and therefore may specifically enhance the functional output within PBs. Consistent with the hypothesis, increasing PHYB abundance enlarged the PB size (Fig. 1d,e)." I agree with this statement ... however, if phyB levels in the nucleoplasm may or may not be increased, what then is the conclusion? Of course an enlarged PB size is consistent with increasing overall phyB levels, but this does not exclude that also the level in the nucleoplasm increases. Yet, according to the authors, the purpose of overexpressing phyB was to specifically disturb PBs without affecting the phyB concentration in the nucleoplasm in order to distinguish between the action of phyB in PBs and the nucleoplasm. If it is unclear (i.e. not experimentally confirmed) that phyB levels do not change in the nucleoplasm in the phyB OX lines, this plan does not work out. Furthermore, in line 485 the authors say "... implying that the nucleoplasmic PHYB increased in gPBC-25 to promote PIF5 degradation.". If I understand this sentence correctly, even the authors argue that phyB levels also increase in the nucleoplasm in overexpression lines and therefore a key assumption for their study is no longer valid.

11. Response to question 1 by reviewer #1: The comment by the reviewer is highly relevant. The reviewer suggests approaches to confirm that the effects observed by the authors are indeed relevant and not artefacts due to overexpression. However, the authors did not do any of these experiments.

Minor comments:

13. Please describe the quantification of western blot data in the methods section (i.e. not only

how the seedlings were grown, etc., but how the bands were quantified). I think this is what reviewer #2 was asking for.

Response to Reviewer #3

We thank Reviewer #3 for the constructive and thoughtful comments. We think that the main concern of Reviewer #3 is about our new strategy of using PHYB overexpression lines in our PB studies. The reviewer questioned whether the observation of PIF5 stabilization could be an artifact that happens only when PHYB is overexpressed but not under natural conditions. Because many questions are related to this concern, it makes it easier to address the particular question first. This question is relevant to two major points we wanted to make in the paper: 1) PB-mediated PIF5 stabilization is likely the long-sought-after mechanism that stabilizes PIF5 in the light, and 2) overexpression lines provide a more effective means, compared to loss-of-function mutants, for assessing the function of PBs (or membraneless organelles in general).

1) PB-mediated PIF5 stabilization is likely the long-sought-after mechanism that stabilizes PIF5 in the light.

As we mentioned in the Introduction and Discussion sections, one reason why we decided to switch to the PIF5 model is that PIF5 can accumulate in prolonged light conditions [Lorrain et al. (2007) *Plant J* 53:312; Qiu et al. (2017) *Nat Commun* 8:1905; Pham et al. (2018) *Plant J* 96(2):260; Sharma et al. (2019) *Nat Commun* 10:4417]. It is widely accepted that PHYB promotes the degradation of PIF1, PIF3, PIF4, and PIF5. However, it remains puzzling why PIF5 (and also PIF4) can accumulate in the light when PHYB is active. This suggests that while PIF5 (and PIF4) are being degraded by PHYB, there has to be a counterbalancing mechanism to stabilize these PIFs under natural conditions [Qiu et al (2021) *Nat Commun* 12:2042]. However, such a stabilizing mechanism remains elusive. Our results here suggest that the function of PBs is to stabilize PIF5 under natural conditions. This conclusion is supported by the following evidence:

1. We showed that PHYB forms PBs under the physiological concentration using immunostaining (Fig. 2D) [Willige et al. (2021) *Nat Genet* 53:955].
2. We demonstrated that PIF5 localizes to PBs when PHYB is in the physiological concentration (Fig. 2D).
3. There was clearly a counterbalancing mechanism between PIF5 stabilization and degradation. We showed that the balance of these two mechanisms can be switched in four scenarios, using *phyB* mutant, wild-type and PHYB overexpression lines.
 - a. Col-0, PIF5 can accumulate – balanced PIF5 stabilization and degradation (Fig. 3).
 - b. *phyB-9*, the balance is switched toward PIF5 stabilization (Fig. 3)
 - c. *gPBC-25*, the balance is switched toward PIF5 degradation (Fig. 3).
 - d. *PBC* (strong light), the balance is switched toward PIF5 stabilization (Fig. 3).
 - e. *PBC* (dim light), the balance is switched back toward PIF5 degradation (Fig. 5).

We used these different lines to perturb the system to demonstrate that the balance of natural occurring PIF5 degradation and stabilization can be rebalanced by manipulating PHYB and PBs. PIF5 accumulation is not only observed in the PHYB overexpression lines. Even within the overexpression lines, we showed that the balance could be switched back and forth depending on light conditions and PB size.

4. PIF5 is stabilized in PBs, this is supported directly by the accumulation of longer-lived mCherry-PIF5 in *PBC* (Fig. 4).

2) Overexpression lines provide a more effective means, compared to loss-of-function mutants, for assessing the function of PBs (or membraneless organelles in general).

We do understand the reviewer's concerns about overexpression lines. In general, to define the function of a gene, evidence generated using loss-of-function mutants is often considered more valuable and convincing than evidence gathered using gain-of-function alleles. For this reason, in the past 20 years, we have identified both intragenic phyB mutants and extragenic mutants required for PB formation. As described in the Introduction section, it took us two decades to finally learn the lesson that loss-of-function PHYB or PHYB-signaling mutants, although they may reveal a correlation between PB assembly and a particular signaling output, cannot uncouple the signaling output of PBs and the surrounding nucleoplasm and, therefore, cannot unequivocally define the function of PBs. This study represents our first attempt to use PHYB overexpression to study PB functions. The outcome of this study demonstrates that, counterintuitively, perturbing PB size by overexpressing PHYB is a more effective approach to understanding the function of PBs (this is likely applicable to other membraneless organelles).

Reviewer #3

Formation of photobodies is an intriguing feature of the plant photoreceptor phytochrome B (phyB). It has been observed the first time more than two decades ago, but their molecular nature and to a large extent also their physiological function remained elusive. In the work presented in this manuscript, the authors perturbed photobody size by overexpression of phyB. They observed that moderately increased levels of phyB lead to destabilisation of PIF5, an important phyB downstream signalling factor, while strong overexpression of phyB stabilises PIF5. The authors interpretation is that PIF5 is stabilised in phyB photobodies, while it is targeted for degradation when localising to the nucleoplasm.

Overall, this is an interesting manuscript with a very surprising result. The experimental data looks solid and clear, and my concerns are mainly related to the interpretation of the data as detailed in my comments below.

1. Introduction, line 112ff: The authors say that degradation of PIF1/3/4/5 is a central mechanism to control their activity by phyB, but they do not mention that binding of phyB to PIF1 and PIF3 (and potentially other PIFs) inhibits their association with target promoters. It has been shown that NGB (N-terminal half of phyB fused to GUS and an NLS) is active in triggering light signalling, but does not promote the degradation of PIF3. Instead, NGB inhibits association of PIF1/3 with promoters, suggesting that light signalling may not require degradation of PIFs. Thus, it is essential to mention this mechanism and to cite the relevant publication by Giltsu Chois lab: Park et al., 2012, Plant J., 72, 537–546.

Response: We thank the reviewer for the comments. We had previously included the references related to PHYB-mediated regulation of PIF activity. However, because the number of references in the original version was over the maximum limit allowed at *Nat Commun*, so we removed them. But, as the reviewer pointed out, we agree it is important to include those important references. We have now added them back and revised the sentence to:

“A central mechanism of PHYB signaling is to inhibit the function of PIFs, including promoting their degradation^{18,40}, attenuating their DNA binding^{41,42}, and repressing their transactivation activity⁴³.”

2. Introduction, line 170ff: I do not understand this sentence: "PHYB condensation stabilizes PIF5 in PBs to counteract PIF5 degradation in the surrounding nucleoplasm, thereby enabling a counterbalancing mechanism to titrate nuclear PIF5 and its signaling outputs.". What does it mean in simple words?

Response: We thank the reviewer for the comments. We revised the sentence to “PHYB condensation stabilizes PIF5 in PBs to counteract PIF5 degradation in the surrounding nucleoplasm, *which* enables a counterbalancing mechanism to titrate nuclear PIF5 and its signaling outputs.”

This means that the PB-dependent PIF5 stabilization enables a counterbalancing mechanism to titrate nucleoplasmic PIF5, because, without such a stabilization mechanism, PIF5 would have been degraded in the light.

3. Introduction, line 172ff: As take home message, the authors write "This novel PB-mediated PHYB signaling mechanism provides a framework for regulating a plethora of PHYB-interacting signaling molecules to generate diverse environmental responses in plants.". I don't think that this very broad and general statement is appropriate, given that even PIFs differ regarding accumulation in light, factors required for degradation, etc. It is also interesting that a previous publication by the authors lab (Fig. 1d in Qiu et al., 2017, Nat. Commun., 8, 1905) has shown that levels of PIF3 reversely correlate with levels of phyB, suggesting that the effect observed for PIF5 may even not be common among PIFs. Furthermore, the effect observed for PIF5 occurs in lines that have phyB levels much higher than in WT plants, i.e. the regulatory mechanism suggested by the authors may exist ... but it may not be relevant under natural conditions.

Response: We thank the reviewer for the comments. We revised the sentence to: “This novel PB-enabled counterbalancing mechanism provides a framework for assessing the function of PBs in regulating other PB constituents in plant light and temperature signaling.”

Regarding the comments on “the effect observed for PIF5”, please see the beginning section.

4. Results, line 206ff: The authors say that phyB levels in photobodies increased with overall phyB levels but there are no data on phyB levels in the nucleoplasm. However, for the conclusions of the manuscript, it is prerequisite that levels in the nucleoplasm do not change (or at least do not increase with increasing overall phyB levels). Thus, demonstrating this is essential for the conclusions of the manuscript.

Response: When comparing PBs in *gPBC-25*, *gPBC-29*, and *PBC* lines (Fig. 1), we measured two parameters: 1) 3D volumes of PBs (Fig. 1e) and 2) PHYB partition to PBs (Fig. 1f). The PB partition of PHYB was calculated as PHYB-CFP signals in PBs vs those in the entire nucleus (PB + nucleoplasm), using a 3-D model. As shown in Fig. 1f, PB partition of PHYB increased from *gPBC-25* to *PBC*, indicating that not only the PB size became larger, the fraction of PHYB to PBs also increased in *PBC*.

5. Results, p. 268ff: *I do not question the data showing that the PIF5 level decreases with a moderate increase in phyB levels, while it increases with a strong overexpression of phyB. However, I doubt that this effect is relevant under natural conditions. Even in the phyB OX line with 8-fold overexpression, PIF5 levels are reduced (and not increased) compared to wild type, i.e. there would have to be natural conditions under which phyB levels increase >8-fold for the effect to be relevant. What are these conditions?*

Response: Please see the opening section. We will explain further how we think about *gPBC-25* below.

6. Results, line 280ff: *The authors write "These results suggest that PHYB signaling exerts opposing roles in both degrading and stabilizing PIF5, the balance of which can be adjusted by altering the PHYB level."  Please give examples for natural conditions (in wildtype plants) under which phyB levels increase more than 8-fold (8-fold overexpression still promotes PIF5 degradation). If such conditions do not exist, the effect of PIF5 stabilisation at strongly increased phyB levels is physiologically not relevant.*

Response: Please see our response in the opening section.

7. Discussion, line 370: *"Because control of the stability of PHYB-associated signaling components is a major mechanism of light signaling, the novel PB-dependent counterbalancing mechanism provides the framework for regulating a plethora of PHYB-interacting signaling molecules in diverse plant environmental responses. W"  A) This mechanism only works if phyB is strongly (>8-fold) overexpressed and therefore may be irrelevant under natural conditions (unless the authors can define natural conditions under which phyB levels are upregulated to the required extent in wildtype seedlings/plants). B) The authors have shown the effect for PIF5, not for any other protein. Moreover, from their own publication (Qiu et al., 2017, Nat. Commun., 8, 1905), there is evidence that this effect may not exist for PIF3. Thus, the sentence in line 370ff should be rephrased.*

Response: We thank the reviewer for the comments. We have revised the sentence to “..., this novel PB-dependent counterbalancing mechanism for PIF5 regulation provides the framework for assessing the function of PBs in regulating other PHYB-interacting signaling molecules in light and temperature signaling.”

8. Discussion, line 408ff: *"Together, our results reveal that in addition to promoting PIF5 degradation, PHYB also stabilizes PIF5 by recruiting PIF5 to PBs, providing a mechanism that allows PIF5 to accumulate in the light."  Please clearly say that this depends on strong overexpression of phyB; even 8-fold overexpression is not sufficient to observe the effect!*

Response: Please see the opening section.

9. Discussion, line 411f: "PHYB-mediated degradation of PIFs has been invoked as the central mechanism of light signaling"  I agree that PIF degradation has the potential to affect light signalling. However, Park et al., 2012, *Plant J.*, 72, 537–546 have shown that *phyB* also controls binding of PIF1 and PIF3 to target promoters and that this mechanism is apparently sufficient to control light signalling in NGB background, where PIF3 is not degraded in light. Furthermore, PIF7 is not primarily regulated by degradation. Thus, the sentence in line 411f is misleading and puts too much emphasis on the importance of degradation, while neglecting regulation of promoter binding. In this context, it is appropriate to mention the work by Park et al., 2012, *Plant J.*, 72, 537–546 and include it as reference.

Response: We agree. Here, because we wanted to focus on PIF degradation, we revised the sentence to "PHYB-mediated degradation of PIFs has been invoked as one the most important mechanisms of light signaling^{18,40}."

10. Discussion, line 471ff: "Here, we perturbed the PB size by increasing PHYB abundance. If PB formation is driven by the LLPS of PHYB, increasing the PHYB level is expected to only enlarge the PB size without changing the concentrations of PHYB in the nucleoplasm and therefore may specifically enhance the functional output within PBs. Consistent with the hypothesis, increasing PHYB abundance enlarged the PB size (Fig. 1d,e)."  I agree with this statement ... however, if *phyB* levels in the nucleoplasm may or may not be increased, what then is the conclusion? Of course an enlarged PB size is consistent with increasing overall *phyB* levels, but this does not exclude that also the level in the nucleoplasm increases. Yet, according to the authors, the purpose of overexpressing *phyB* was to specifically disturb PBs without affecting the *phyB* concentration in the nucleoplasm in order to distinguish between the action of *phyB* in PBs and the nucleoplasm. If it is unclear (i.e. not experimentally confirmed) that *phyB* levels do not change in the nucleoplasm in the *phyB* OX lines, this plan does not work out. Furthermore, in line 485 the authors say "... implying that the nucleoplasmic PHYB increased in *gPBC-25* to promote PIF5 degradation.". If I understand this sentence correctly, even the authors argue that *phyB* levels also increase in the nucleoplasm in overexpression lines and therefore a key assumption for their study is no longer valid.

Response: These are interesting questions. First, we agree with the reviewer that the biphasic PIF5 responses suggest that the nucleoplasmic PHYB level must have changed when PHYB abundance was increased in the overexpression lines. The LLPS theory is an *in vitro* model (there is no *in vivo* LLPS theory) – which is why it is extremely valuable to test this model in the PB system *in vivo*. The biphasic PIF5 response, especially the switch to PIF5 degradation from Col-0 to *gPBC-25*, suggests that the initial increase in PHYB level in *gPBC-25* must have increased the nucleoplasmic PHYB, which promotes PIF5 degradation. One possibility is that the LLPS of PHYB *in vivo* can occur below the critical concentration required for global LLPS. In that case, increasing PHYB initially in *gPBC-25* should also increase the concentration of the nucleoplasmic PHYB. So, what enables PHYB to phase separate below the critical concentration *in vivo*? – This is a general question that the entire LLPS field needs to explain when applying this *in vitro* framework to live cells. Our results suggest that a simple LLPS model cannot explain our observations. We stated, "One possibility is that at the physiological PHYB concentration, PHYB condensation may require a complex mechanism beyond LLPS."

11. Response to question 1 by reviewer #1: The comment by the reviewer is highly relevant. The reviewer suggests approaches to confirm that the effects observed by the authors are indeed relevant and not artefacts due to overexpression. However, the authors did not do any of these experiments.

Response: Please see our responses in the beginning section.

Minor comments:

13. Please describe the quantification of western blot data in the methods section (i.e. not only how the seedlings were grown, etc., but how the bands were quantified). I think this is what reviewer #2 was asking for.

Response: We revised the Method section.

Reviewer #3 (Remarks to the Author):

I thank the authors for their responses.

I understand the purpose of using the phyB OX lines and I agree with the interpretation that high overexpression stabilises PIF5. I also agree that the mechanism as suggested by the authors exists. However, I am not convinced that this mechanism is relevant under natural conditions which is what the authors claim. In a range of phyB levels from absent (phyB mutant) to WT levels (Col-0) to 8x overexpression (gPBC-25), there is a correlation between phyB levels and degradation of PIF5. Only at higher than 8x overexpression, the mechanism suggested by the authors becomes relevant and PIF5 is stabilised instead of destabilised by phyB. My question was if there are any natural conditions where there are more than 8x increased phyB levels compared to WT. The authors did not answer this question. If such conditions do not exist, the effect described in the manuscript is not relevant under natural conditions (even if it exists under artificial conditions). The authors argue that even in the phyB OX lines the system can be switched from PIF5 stabilisation to PIF5 degradation by changing the light conditions (i.e. inactivating phyB). However, this still does not answer my question.

In response to point 3 of my previous comments, the authors rephrased the respective sentence to "This novel PB-enabled counterbalancing mechanism provides a framework for assessing the function of PBs in regulating other PB constituents in plant light and temperature signaling.". Although it cannot be excluded that other components are regulated in a similar manner as PIF5, I still wonder how likely this is given that even PIF3 is not regulated by this mechanism. Regarding this, the authors did not comment on their previous finding that PIF3 is not stabilised by phyB OX.

Regarding levels of phyB in the nucleoplasm, the authors argue that the fraction of phyB in PBs increases with increasing total phyB levels. I agree with this, but this does not exclude that also levels in the nucleoplasm increase (just not to the same extent as the levels in PBs). Furthermore, the authors say in their response letter "The biphasic PIF5 response, especially the switch to PIF5 degradation from Col-0 to gPBC-25, suggests that the initial increase in PHYB level in gPBC-25 must have increased the nucleoplasmic PHYB, which promotes PIF5 degradation.". Thus, they essentially agree that in the range from no phyB to 8x overexpression of phyB, PIF5 degradation correlates with phyB levels. Only at higher overexpression of phyB, PIF5 is stabilised. I am not convinced that this effect is relevant under natural conditions.

Response to Reviewer #3

Reviewer #3 (Remarks to the Author):

I thank the authors for their responses.

I understand the purpose of using the phyB OX lines and I agree with the interpretation that high overexpression stabilises PIF5. I also agree that the mechanism as suggested by the authors exists. However, I am not convinced that this mechanism is relevant under natural conditions which is what the authors claim. In a range of phyB levels from absent (phyB mutant) to WT levels (Col-0) to 8x overexpression (gPBC-25), there is a correlation between phyB levels and degradation of PIF5. Only at higher than 8x overexpression, the mechanism suggested by the authors becomes relevant and PIF5 is stabilised instead of destabilised by phyB. My question was if there are any natural conditions where there are more than 8x increased phyB levels compared to WT. The authors did not answer this question. If such conditions do not exist, the effect described in the manuscript is not relevant under natural conditions (even if it exists under artificial conditions). The authors argue that even in the phyB OX lines the system can be switched from PIF5 stabilisation to PIF5 degradation by changing the light conditions (i.e. inactivating phyB). However, this still does not answer my question.

PIF5 can also accumulate in Col-0 – without a stabilization mechanism, it would be degraded by PHYB signaling. Supporting this model, increasing PHYB in gPBC-25 enhances PIF5 degradation.

Answering the question regarding PHYB levels in natural conditions would require a thorough examination of PHYB levels in all natural conditions – this is no answer to this question. The reviewer argues that if the perturbation condition used in a study does not occur naturally, the resulting conclusion would be irrelevant to the natural condition. In our opinion, this argument is invalid, because, applying the same argument, data generated using gene deletion mutants would also be irrelevant because most gene deletion mutants do not exist in nature.

With that said, we appreciate the reviewer's comments. We added the following in the Discussion section: "Although we could not have revealed the function of PBs in stabilizing PIF5 without perturbing PBs using the PHYB overexpressing lines, it is important to note that a limitation of characterizing PB dynamics using PHYB overexpression lines would be the difficulty of completely excluding the possibility of a secondary effect on the signaling output due to PHYB overexpression."

In response to point 3 of my previous comments, the authors rephrased the respective sentence to "This novel PB-enabled counterbalancing mechanism provides a framework for assessing the function of PBs in regulating other PB constituents in plant light and temperature signaling.". Although it cannot be excluded that other components are regulated in a similar manner as

PIF5, I still wonder how likely this is given that even PIF3 is not regulated by this mechanism. Regarding this, the authors did not comment on their previous finding that PIF3 is not stabilised by phyB OX.

As stated in the Discussion section, the function of PBs for each PIF should be studied individually. The data here does not suggest whether PIF3 is or is not regulated by the same mechanism as PIF5. We stated clearly in the Introduction section that our previously published data on PIF3 using loss-of-function PHYB or PHYB signaling mutants only showed a correlation between PBs and PIF3 degradation but could not unequivocally conclude whether the PBs are the sites for PIF3 degradation.

Regarding levels of phyB in the nucleoplasm, the authors argue that the fraction of phyB in PBs increases with increasing total phyB levels. I agree with this, but this does not exclude that also levels in the nucleoplasm increase (just not to the same extent as the levels in PBs). Furthermore, the authors say in their response letter "The biphasic PIF5 response, especially the switch to PIF5 degradation from Col-0 to gPBC-25, suggests that the initial increase in PHYB level in gPBC-25 must have increased the nucleoplasmic PHYB, which promotes PIF5 degradation.". Thus, they essentially agree that in the range from no phyB to 8x overexpression of phyB, PIF5 degradation correlates with phyB levels. Only at higher overexpression of phyB, PIF5 is stabilised. I am not convinced that this effect is relevant under natural conditions.

We have addressed this question in our previous response. If PB formation had followed the simple model of LLPS, we would expect to observe a linear PIF5 response. Apparently, a simple LLPS model cannot explain the unexpected biphasic PIF5 response. As stated in the Discussion section, "One possibility is that at the physiological PHYB concentration, PHYB condensation may require a complex mechanism beyond LLPS."